# Mind's Eye of LLMs: Visualization-of-Thought Elicits Spatial Reasoning in Large Language Models

**Wenshan Wu**[†]   **Shaoguang Mao**[†]   **Yadong Zhang**[†, ‡, *]
**Yan Xia**[†]   **Li Dong**[†]   **Lei Cui**[†]   **Furu Wei**[†]
[†]Microsoft Research   [‡]East China Normal University

## Abstract

Large language models (LLMs) have exhibited impressive performance in language comprehension and various reasoning tasks. However, their abilities in spatial reasoning, a crucial aspect of human cognition, remain relatively unexplored. Human possess a remarkable ability to create mental images of unseen objects and actions through a process known as **the Mind's Eye**, enabling the imagination of the unseen world. Inspired by this cognitive capacity, we propose Visualization-of-Thought (**VoT**) prompting. VoT aims to elicit spatial reasoning of LLMs by visualizing their reasoning traces, thereby guiding subsequent reasoning steps. We employed VoT for multi-hop spatial reasoning tasks, including natural language navigation, visual navigation, and visual tiling in 2D grid worlds. Experimental results demonstrated that VoT significantly enhances the spatial reasoning abilities of LLMs. Notably, VoT outperformed existing multimodal large language models (MLLMs) in these tasks. While VoT works surprisingly well on LLMs, the ability to generate *mental images* to facilitate spatial reasoning resembles the mind's eye process, suggesting its potential viability in MLLMs. Please find the dataset and codes in our [project page](#).

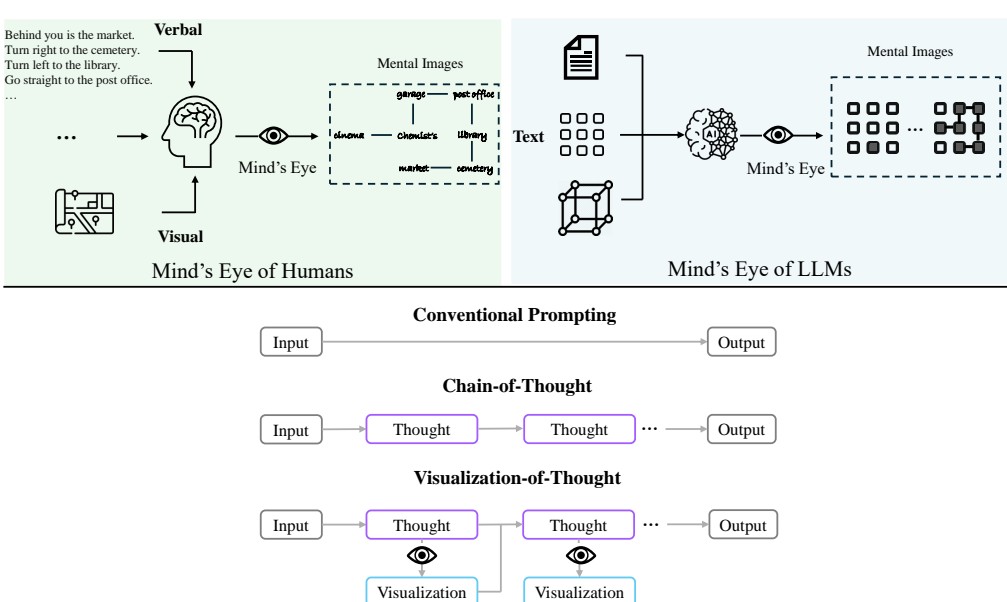

Figure 1: Humans can enhance their spatial awareness and inform decisions by creating mental images during the spatial reasoning process. Similarly, large language models (LLMs) can create internal *mental images*. We propose the VoT prompting to elicit the "mind's eye" of LLMs for spatial reasoning by visualizing their thoughts at each intermediate step.

---

*Contribution during internship at Microsoft Research.

38th Conference on Neural Information Processing Systems (NeurIPS 2024).

# 1 Introduction

Recently, large language models (LLMs) [BCE⁺23, BMR⁺20, TLI⁺23, JSM⁺23] have achieved remarkable performance on various language-related tasks. However, despite their success in math reasoning [KGR⁺23], common sense reasoning [LKH⁺22], and other reasoning tasks such as symbolic reasoning or logic reasoning [KGR⁺23], their abilities in spatial reasoning still remain underexplored [RFD⁺21, YBL⁺23, MHV⁺24].

Spatial reasoning is an essential function of human cognition, allowing us to interact with the environment. It facilitates tasks that require understanding and reasoning about the spatial relationships between objects and their motions. The spatial reasoning of language models largely relies on language to reason about spatial information, whereas human cognitive capabilities extend far beyond verbal reasoning. Humans can not only create task-relevant abstract representations from visual perception [BK18, KC22], but also imagine unseen scenes through their *mind's eye*. It remains a research topic called mental image [She78] in domains of neuroscience, philosophy of mind, and cognitive science. Building upon this cognitive function, humans facilitate spatial reasoning by mental image manipulation, such as navigation [Tol48], mental rotation [SM71], mental paper folding [SF72], and mental simulation [MK09]. Figure 1 illustrates the human process involved in a navigation task. Humans enhance their spatial awareness and inform their decisions by creating mental images of a route, utilizing various sensory inputs such as navigation instructions or a map image. Subsequently, they simulate route planning through the mind's eye.

Inspired by this cognitive mechanism, we conjecture that LLMs possess the ability to create and manipulate *mental images* in the mind's eye for spatial reasoning. As illustrated in Figure 1, LLMs could potentially process and understand spatial information in various formats. They might be capable of visualizing internal states and manipulating these *mental images* through their *mind's eye*, thereby guiding subsequent reasoning steps to enhance spatial reasoning. Therefore, we propose the **Visualization-of-Thought (VoT)** prompting to elicit this ability. This method leverage LLMs to visualize their reasoning steps and inform subsequent steps, implementing the concept of visuospatial sketchpad [Bad92]. VoT adopts zero-shot prompting instead of relying on few-shot demonstrations or text-to-image visualization with CLIP [RKH⁺21]. This choice stems from LLMs' ability to acquire various *mental images* from text-based visual art [SB14, SMM21, Reg19].

To evaluate the effectiveness of **VoT** in spatial reasoning, we selected three tasks that require spatial awareness in LLMs, including natural-language navigation [YBL⁺23], visual navigation, and visual tiling. These tasks require an understanding of space, direction, and geometric shape reasoning. To emulate human-like multisensory perception, we designed 2D grid worlds using special characters as enriched input formats for the LLMs in visual navigation and visual tiling tasks. We compared different models (GPT-4, GPT-4V) and prompting techniques across these three tasks. The findings reveal that the VoT prompting proposed in this paper consistently induces LLMs to visualize their reasoning steps and inform subsequent steps. Consequently, this approach achieved significant performance improvements on the corresponding tasks.

The main contributions of this paper include:

**1**. We shed light on LLMs' *mental image* for spatial reasoning from a cognitive perspective, conducting quantitative and qualitative analyses on the mind's eye of LLMs and its limitations. We also explore cues about the origin of this generalized ability from code pre-training.

**2**. We develop two tasks of "visual navigation" and "visual tiling", along with corresponding synthetic datasets, emulating various sensory inputs for LLMs. These tasks are structured to support varying levels of difficulty, offering a well-designed testbed for the research on spatial reasoning.

**3**. We propose **Visualization-of-Thought** (**VoT**) prompting to elicit the mind's eye of LLMs for spatial reasoning and provide empirical evaluations on three tasks. Experiment results prove the effectiveness of VoT prompting compared with other prompting methods and existing MLLMs. This ability to generate *mental images* to facilitate spatial reasoning resembles the mind's eye process, suggesting its potential viability in MLLMs.

## 2 Spatial Reasoning

Spatial reasoning refers to the ability to comprehend and reason about the spatial relationships among objects, their movements, and interactions with the environment. This skill is vital for a wide range of real-world applications such as navigation, robotics, and autonomous driving. These fields necessitate action planning based on visual perception and a concrete understanding of spatial dimensions.

Although several tasks and datasets [WBC+15, SZL22, MK22, LB18, RAB+20] have been developed to probe the spatial semantics embedded in text, existing research efforts often focus on how spatial terms are linguistically structured. Recently, significant achievements and impressive results have been achieved in these benchmarks by converting spatial terms to logical forms through LLMs and adopting logic programming [YIL23]. This implies that excelling in these tasks does not necessarily equate to a genuine understanding of spatial information by LLMs, nor does it provide an accurate measure of their spatial awareness.

Spatial awareness involves understanding spatial relationships, directions, distances, and geometric shapes, all of which are essential for action planning in the physical world. To evaluate the spatial awareness and spatial reasoning abilities of LLMs, we have selected tasks that test navigation and geometric reasoning skills, including natural language navigation, visual navigation and visual tiling.

### 2.1 Natural Language Navigation

Natural language navigation task [YBL+23] was inspired by prior research on human cognition [GDB17] presenting participants with sequential transitions sampled from a graph structure.

In this context, a square map is defined by a sequence of random walk instructions and associated objects at each location, denoted as $W = \{(l_1, o_1), (l_2, o_2), \ldots, (l_n, o_n)\}$. Given a square map $W$, and sequence of navigation instructions $I = \{i_1, \ldots, i_k\}$, the task for the model is to identify the associated object $o \in W$ at the specified location $l$ which is determined by the navigation instructions, as detailed in Equation 1 and exemplified in Appendix B.2.

$$o \sim p(o \in W | W = \{(l_1, o_1), (l_2, o_2), \ldots, (l_n, o_n)\}, I) \tag{1}$$

### 2.2 Visual Navigation

Visual navigation task presents a synthetic 2D grid world to LLM, challenging it to navigate using visual cues. The model must generate navigation instructions to move in four directions (left, right, up, down) to reach the destination from the starting point while avoiding obstacles. This involves two sub-tasks: **route planning** and **next step prediction**, requiring multi-hop spatial reasoning, while the former is more complex. Task instructions are available in Figure 6 in appendix.

**Formulation** The model is presented with a grid map $M$ consisting of $k$ consecutive edges $E = \{e(s_0, s_1), e(s_1, s_2), \cdots, e(s_{k-1}, s_k)\}$, where the starting point and destination are $s_0$ and $s_k$ respectively, as shown in Figure 2. Route planning task is to generate a sequence of correct directions $D = \{d(s_0, s_1), d(s_1, s_2), \cdots, d(s_{k-1}, s_k)\}$, as defined in Equation 2. Given $M$ and $t$ navigation instructions $D_{t,0<t<k} = \{d(s_0, s_1), \cdots, d(s_{t-1}, s_t)\}$, next step prediction task is to identify the correct direction $d(s_t, s_{t+1})$ of the next step, as defined in Equation 3.

$$D \sim p(\{d(s_0, s_1), d(s_1, s_2), \cdots, d(s_{k-1}, s_k)\} \mid M) \tag{2}$$

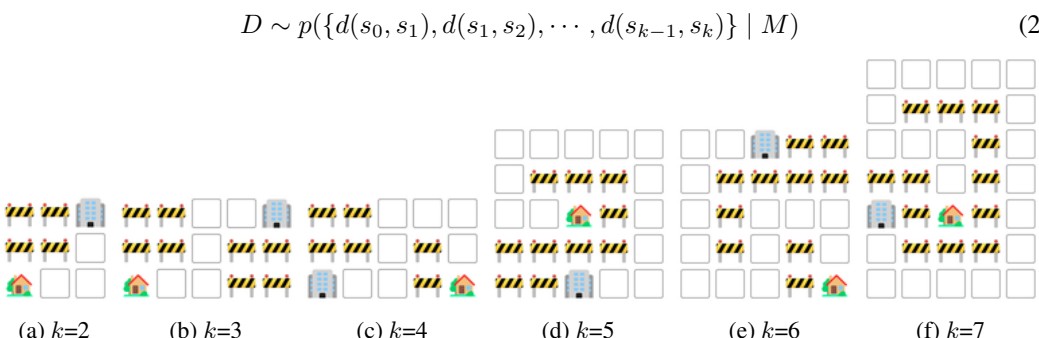

(a) $k$=2  (b) $k$=3  (c) $k$=4  (d) $k$=5  (e) $k$=6  (f) $k$=7

Figure 2: Examples of a navigation map under different settings of $k$, with emoji of house indicating the starting point, and emoji of office indicating the destination.

$$d \sim p(d(s_t, s_{t+1}) \mid M, D_{t,0<t<k}) \tag{3}$$

**Implementation**   The navigation map's underlying graph is semi-Eulerian, alternating between horizontal and vertical edges, with $2^{k+1}$ possible spatial configurations for a $k$-hop navigation map. For each map and set of $k$ navigation instructions, $k - 1$ question-and-answer (QA) instances,i.e. "what is the next step?" are created. Further implementation details are in Appendix A.1.

## 2.3   Visual Tiling

Introduced by [Gol66], polyomino tiling is a classic spatial reasoning challenge. We extend this concept to test the LLM's ability to comprehend, organize, and reason with shapes in a confined area, thus enhancing the evaluation of spatial reasoning skills. As depicted in Figure 3, the task involves a rectangle with unfilled cells and various polyomino pieces, like the I-tetromino made of four aligned squares. The model must select the appropriate polyomino variant, such as choosing the orientation for the I-tetromino, to solve the QA puzzle. Task instructions are provided in Figure 7 in appendix.

**Formulation**   The model is presented with a rectangle $R$ masked with $k$ unique polyominoes $MP = \{mp_1, \cdots, mp_k\}$, 2 corresponding variants of each polyomino $v_{i<=k} = \{v_{i1}, v_{i2}\}$, and a polyomino query $q \in MP$. Visual tiling task is to identify the correct variant of $q$, as defined in Equation 4.

$$v \sim p(v_q \mid R, \{mp_1, \cdots, mp_k\}, \{v_{11}, v_{12} \cdots, v_{k1}, v_{k2}\}, q) \tag{4}$$

**Implementation**   The dataset comprises valid spatial arrangements generated through existing algorithms[ES03, GN07], with random masking of polyominoes to create QA puzzles. Details are provided in Appendix A.2.

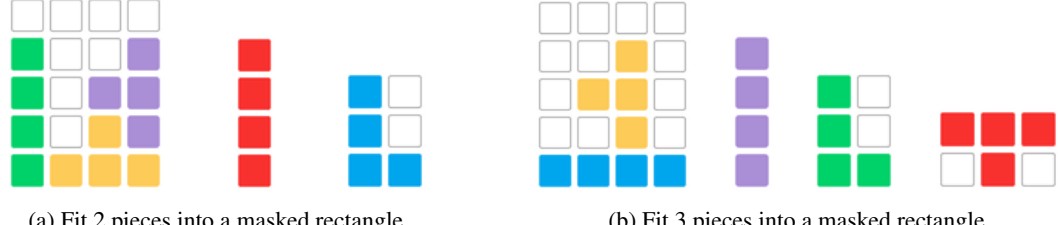

    (a) Fit 2 pieces into a masked rectangle                   (b) Fit 3 pieces into a masked rectangle

Figure 3: Example of visual tiling with masked polyomino pieces. Variants of those polyomino pieces including rotation and reflection are not shown in this figure.

## 3   Visualization-of-Thought Prompting

Considering the way humans process spatial information during tasks like navigation, it's common to create mental images , such as maps, to enhance spatial awareness or simulating movements to inform decision-making. Our objective is to elicit the spatial awareness of LLMs and ground their reasoning by visualizing the consequence of their intermediate reasoning steps.

We introduce **Visualization-of-Thought (VoT)** prompting: **"Visualize the state after each reasoning step."** This new paradigm for spatial reasoning aims to generate reasoning traces and visualizations in an interleaved manner. Qualitative results of this approach are presented in Figure 4.

We use $p_\theta$ to denote a pre-trained LM with parameters $\theta$, $x, y, z$ to denote a language sequence, and $v$ to denote a visualization sequence in text form. In a multi-hop spatial reasoning task with input $x$, CoT prompting generates a series of intermediate steps $z_1, \cdots, z_n$, each step $z_i \sim p_\theta(z_i \mid x, z_{1\ldots i-1})$ is sampled sequentially, followed by the output $y \sim p_\theta(y|x, z_{1\cdots n})$. As shown in Figure 1, **VoT** prompting enhances this process by adding a visuospatial sketchpad to each intermediate step $z_i$, then the subsequent step $z_{i+1}$ is sampled conditioned on prior steps $z_{1\ldots i}$ and visualizations $v_{1\ldots i}$.

As defined in the Equation 5 and 6, it forms interleaved reasoning traces and visualizations. A qualitative comparison between outputs of VoT and CoT is provided in Figure 8a in appendix.

$$v_i \sim p_\theta(v_i \mid prompt_{VoT}, x, z_{1\ldots i}, v_{1\ldots i-1}) \tag{5}$$

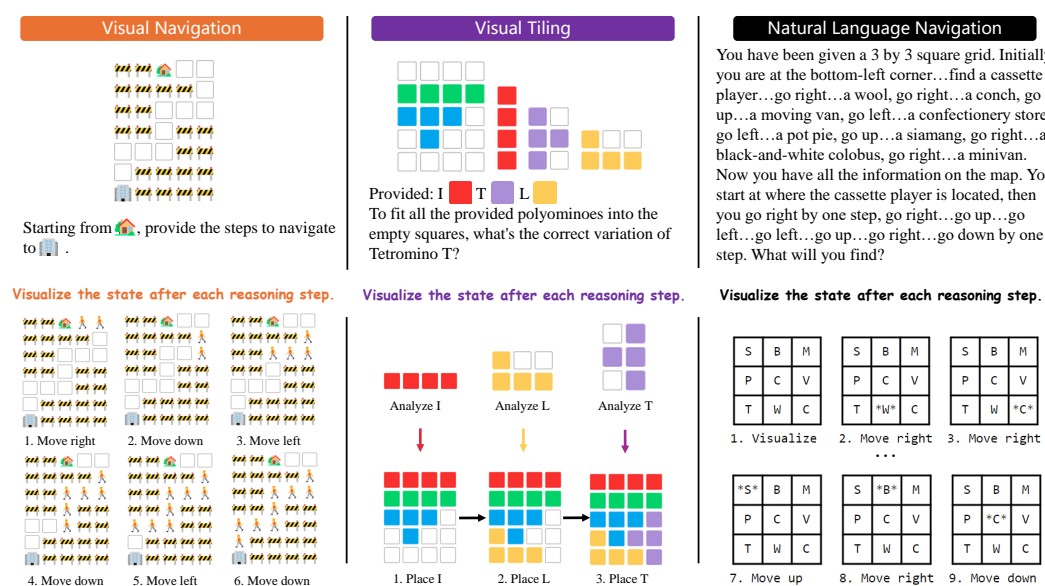

Figure 4: Examples of VoT prompting in three tasks, where LLM generates 2D grids as text-form *mental images*. The generated reasoning traces and visualizations form an interleaved sequence to track the state over time. The 2D grids in the input and responses are composed of special characters. Full responses could be found in Appendix B.

$$z_{i+1} \sim p_\theta(z_{i+1} \mid prompt_{VoT}, x, z_{1...i}, v_{1...i}) \tag{6}$$

This reasoning paradigm enables LLMs with visual state tracking. We introduce the concept of a **state**, denoted as $s_i = [x, z_{1...i}, v_{1...i-1}]$ representing a partial solution at step $i$ with the input, the sequence of intermediate steps $z_{1...i}$ and the sequence of visualizations $v_{1...i-1}$.

$$\begin{aligned} v_i &\sim p_\theta(v_i \mid prompt_{VoT}, x, z_{1...i}, v_{1...i-1}) \\ &\sim p_\theta(v_i \mid prompt_{VoT}, s_i) \end{aligned} \tag{7}$$

As shown in Equation 7, visual state tracking is implemented by generating the visualization $v_i$ as representation of the internal state $s_i$ after each reasoning step $z_i$ (e.g. $v_i$ could be a grid of the navigation map marked with path or a filled rectangle). Grounded by the visual state tracking sequence, the subsequent state is derived by $s_{i+1} \sim p_\theta(s_{i+1} \mid prompt_{VoT}, x, s_i, v_i)$. This mechanism allows for the derivation of subsequent states, reflecting spatiotemporal causality and enhancing the spatial reasoning capabilities of LLMs in a grounded context.

## 4 Experiment

### 4.1 Setup

For the visual tasks where a counterpart image exists for each text input, we conduct additional experiments with a multimodal model. Specifically, we adopt GPT-4 [OA+23] and GPT-4 Vision [Ope23] via Azure OpenAI API as they're state of the art LLM and multimodal model respectively. API settings are *temperature* 0 as greedy decoding and *top p* 1, with model versions of 1106-preview and vision-preview. For all experiments we adopt **zero-shot** prompting.

Depending on whether the LLM is explicitly prompted to visualize intermediate steps, we experiment with three settings of GPT-4, including zero-shot CoT prompting(**GPT-4 CoT**), **GPT-4 w/o Viz** where visualization is explicitly disabled during reasoning, and VoT prompting (**GPT-4 VoT**). Additional setting of GPT-4 Vision with counterpart image input is **GPT-4V CoT**. Prompts are as following:

- **GPT-4 CoT**: Let's think step by step.
- **GPT-4 w/o Viz**: Don't use visualization. Let's think step by step.
- **GPT-4V CoT**: Let's think step by step.
- **GPT-4 VoT**: Visualize the state after each reasoning step.

Task instructions and examples could be found in Appendix B.

## 4.2 Dataset

**Natural Language Navigation**    We generate 200 square maps of size 3x3 which is described by 9 landmarks in snake order traversal, and a set of navigation instructions.

**Visual Navigation**    We generate 496 navigation maps and 2520 QA instances in total, covering various map sizes, up to 7×9 and 9×7. The data distribution is provided in Table 4 in appendix.

**Visual Tiling**    We first generate multiple unique configurations to fill a 5 x 4 rectangle with 5 polyomino pieces including two I tetrominoes, two T tetrominoes and one L tetromino. Then we randomly masked two or three pieces of different types and generate QA instance for each masked pieces. The total number of QA instances is 796, and we show dataset details in Table 5 in appendix.

## 4.3 Metric

We extract the answer from model output by pattern matching. For tasks except for route planning, we calculate accuracy by Equation 8. We adopted sub-string matching[†] as $f_{correct}$ to determine correctness.

$$acc = \sum_i^n f_{correct}(extracted\_answer, ground\_truth)/n \tag{8}$$

For the route planning task which predicts a sequence of navigation instructions, we reject any sequences exceeding 100 instructions, considering them to be random guesses. We then normalize the navigation instructions by executing each navigation instruction. Those instructions which violate navigation rules will be ignored. The length $t$ of normalized instruction sequence is considered as the temporal distance against the starting point. Given the ground-truth of $k$ navigation instructions, the completing rate of route planning is $t/k$. For the dataset of $n$ maps, we report two metrics including:

1. Average completing rate: $\sum_i^n t_i/k_i/n$. Average completing rate among all instruction sequences, reflecting LLM's effectiveness of route planning.
2. Success rate: $\sum_i^n (t_i == k_i)/n$. This metric represents the proportion of instruction sequences with $t = k$, i.e., reaching the destination.

## 4.4 Results

As illustrated in Table 1, **GPT-4 VoT** significantly outperforms other settings in all tasks across all metrics. The significant gap when comparing GPT-4 VoT with GPT-4V CoT and GPT-4 w/o Viz demonstrates that effectiveness of visual state tracking, which allows LLMs visually interpret their actions within an grounded world. And in the natural language navigation task, **GPT-4 VoT** outperforms **GPT-4 w/o Viz** by 23.5%. In the visual tasks, the noticeable performance gap between **GPT-4 CoT** and **GPT-4V CoT** indicates that LLM grounded with 2D grid could possibly outperform a MLLM in challenging spatial reasoning tasks.

On the other hand, performance of GPT-4 VoT is still far from perfect in all tasks, especially in the most challenging route planning task. Despite these tasks are relatively easy for humans, performance of LLMs drops significantly as task difficulty increases. Details on performance trends across difficulty levels are provided in figure 9 and table 6 in appendix.

---

[†]We use this term for simplicity. In natural language navigation tasks, LLMs often output additional words in the extracted answer besides the expected object name. For example, "Answer: You will find ...". In this case, sub-string matching is adopted without affecting the correctness. Otherwise, exact matching is adopted for multiple choice questions in visual tasks.

| Settings | Visual Navigation | | | Visual Tiling | Natural-Language Navigation |
| --- | --- | --- | --- | --- | --- |
| | Route Planning | | Next Step Prediction | | |
| | Completing Rate | Succ Rate | | | |
| GPT-4 CoT | 37.02 | 9.48 | 48.61 | 54.15 | 54.00 |
| GPT-4 w/o Viz | 37.17 | 10.28 | 48.49 | 46.98 | 35.50 |
| GPT-4V CoT | 33.36 | 5.65 | 46.59 | 49.62 | / |
| GPT-4 VoT | **40.77** | **14.72** | **55.28** | **63.94** | **59.00** |

Table 1: Performance of different GPT-4/4V settings in all tasks. Underline denotes statistical significance with $p < 0.05$ when comparing GPT-4 VoT against all baselines using two-sample z-test, while $p < 0.16$ is observed compared with GPT-4 CoT in natural language navigation task.

## 5 Analysis

As explained in section 3, one of the core aspects of VoT lies in enabling LLMs with visual state tracking. During the experiments, it was observed that GPT-4 CoT occasionally exhibited this reasoning pattern across several tasks with exception of route planning. Besides, incorrect visualizations of VoT are commonly observed in model outputs. In this section, our analysis of VoT primarily focuses on three questions: (1) Do visual state tracking behaviors differ among prompting methods? (2) How visualizations enhance final answers? (3) Can VoT benefit less powerful language models?

### 5.1 Do visual state tracking behaviors differ among prompting methods?

For each model output, we extract the sequence of visualizations sampled prior to generating the final answer and discard any visualizations generated thereafter. Then we compare the sequence length $l_v$ with the number of reasoning steps $l_s$. We calculate `Complete Tracking` $\sum_i^n (l_v == l_s)/n$ when a visualization $v_i$ corresponds to each state $s_i$. Similarly, we calculate the `Partial Tracking` metric as $\sum_i^n (l_v > 0)/n$ when at least one visualization is present before the final answer is generated. Figure 5 shows the significant differences between these settings. In the GPT-4 CoT setting, it demonstrated noticeable tracking rate across almost all tasks except route planning. This observation implies that **LLMs inherently exhibit the capability of visual state tracking when spatiotemporal simulation is integral to reasoning**.

On the other hand, the visual state tracking behavior is **sensitive to prompts** to varying degrees. As showcased in Figure 8 in appendix, after removing "reasoning" from the prompt of VoT, the visualizations are sampled after GPT-4 generates the wrong answer. Consequently, explicitly prompting LLMs to visualize their reasoning traces with **VoT markedly improves the visual tracking rate**, thereby enhancing overall performance. The potential contribution of code pre-training to this emergent capability is further explored in Appendix C.

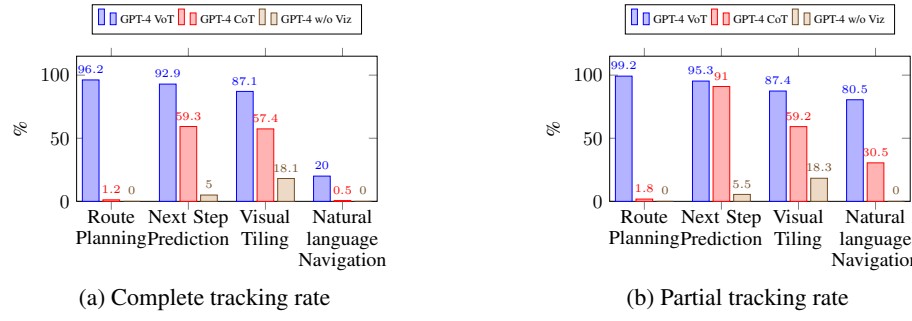

(a) Complete tracking rate        (b) Partial tracking rate

Figure 5: tracking rate of different settings across all tasksk.

### 5.2 How visualizations enhance final answers?

Ideally, **VoT** is supposed to generate an accurate visualization $v_i$ at each step, so that subsequent step $z_{i+1}$ could be determined correctly. This relies on the spatial visualization and spatial understanding capability of LLMs. To evaluate these capabilities of LLMs in these tasks, we extract the final visualization from each model output under the setting **GPT-4 VoT** in visual navigation and polyomino

tiling task. Specifically, for visual navigation task, we extract the visualized map where LLM completed all navigation instructions. For polyomino tiling, we extract the rectangle filled with corresponding polyomino piece. The spatial visualization capability is measured by two criteria: (1) **Compliance**, indicating whether the manipulation of *mental image* satisfies requirements such as avoiding overlap and navigating around obstacles. (2) **Accuracy**, indicating whether the *mental image* aligns with the corresponding state. The spatial understanding capability is measured by the proportion of correct answers when the corresponding visualization is generated accurately.

As could be seen from Table 2, LLMs demonstrate promising potential in performing multi-hop visualization while adhering to spatial constraints, with compliance rates of approximately 51-52%. However, the relatively low accuracy of state visualization (around 24%-26%) indicates a need for significant improvements in this area. Despite this limitation, **LLMs are able to make correct decisions in 65%-77% of the cases when accurate internal state visualizations are generated**, which enhances groundedness and contributes to notable performance gains. Several case studies are provided in Appendix E for interested readers.

| Task | Spatial Visualization | | Spatial Understanding |
|------|-----------|----------|----------|
| | Compliance | Accuracy | Accuracy |
| Visual Navigation | 51.14 | **26.48** | 65.16 |
| Visual Tilling | **52.01** | 24.25 | **77.20** |

Table 2: Spatial visualization/understanding evaluation in visual navigation and visual tiling task.

On the other hand, **VoT prompting might underperform in those tasks where LLMs can leverage logical reasoning without visualizing internal states**. We conducted experiments in natural language navigation within a ring [YBL+23], where navigation instructions are either clockwise or counter-clockwise movements. By normalizing each instruction to a signed number, GPT-4 converts this task to mathematical calculation of adding and modulus operation. For example, instructions of 15 steps clockwise and 3 steps counter-clockwise are normalized to (15 - 3) % 12. Results show that GPT-4 CoT outperforms GPT-4 VoT with 52.5% VS 49.5% among 200 test instances with ring size of 12.

### 5.3 Can VoT benefit less powerful language models?

To evaluate the efficacy of VoT on less powerful language models, we conducted experiments across various model families [BMR+20, OA+23, TLI+23] and model sizes, including **GPT-3.5 turbo**, **LLAMA3-8B-Instruct** and **LLAMA3-70B-Instruct**. We access GPT-3.5 via Azure OpenAI API with model version 1106-preview and apply greedy decoding to all models.

As shown in Table 3, within the same model family, performance improves across all tasks with increases in model size. **LLAMA3-70B VoT** significantly outperforms the baseline across all tasks except for visual tiling, where it aligns closely with results observed in GPT-4. This consistency suggests that VoT offers a scaling advantage when applied to more advanced models, markedly enhancing performance in larger models. In contrast, less capable models tend to rely on random guessing, especially in spatial reasoning tasks. For instance, in the route planning task, GPT-3.5 CoT often resorts to speculative responses, random guessing in nearly half of the instances, which leads to exhaustion of output tokens. While GPT-3.5 VoT effectively minimizes random guesses, such occurrences become increasingly rare with GPT-4 CoT as the model size expands. On the other hand, the reliance on random guessing introduces unpredictability in performance trends for less powerful models. It suggests their limitations in sustaining reliable reasoning processes across different difficulty levels. Details on performance trends are provided in Appendix D.

## 6 Related Works

**Spatial Reasoning over Text**  Spatial reasoning and spatial language understanding [KPM20] in NLP domain mainly focus on semantic representation [CBGG97, Bat10, HK11], spatial information extraction [RMK18, KVOM11], learning and reasoning [KM15, SLYA17, KFP19]. Recent advancements have further explored spatial reasoning within the context of large language models (LLMs). To improve multi-hop spatial reasoning skills of language models, several works [MFNK21, MK22] proposed to pretrain language models with synthetic datasets. An increasing number of dataset were then developed to covers various type of spatial relations in 2D visual scenes [WBC+15, SZL22],

| Settings | Visual Navigation | | | Visual Tiling | Natural-Language Navigation |
| --- | --- | --- | --- | --- | --- |
| | Route Planning | | Next Step Prediction | | |
| | Completing Rate | Succ Rate | | | |
| GPT-3.5 CoT | 16.10 | **2.62** | **17.42** | 44.10 | 8.50 |
| GPT-3.5 VoT | **19.02** | 1.61 | 13.10 | **47.99** | **9.00** |
| LLAMA3-8B CoT | 4.65 | 0 | **28.73** | **47.24** | **16.50** |
| LLAMA3-8B VoT | **4.97** | **0.2** | 26.75 | 46.73 | 15.50 |
| LLAMA3-70B CoT | 19.90 | 2.62 | 49.01 | **56.41** | 26.00 |
| LLAMA3-70B VoT | **30.24** | **5.85** | **54.09** | 56.03 | **32.50** |

Table 3: Performance of VoT in GPT-3.5 and LLAMA3 models. Underline denotes statistical significance with p < 0.05 compared to corresponding CoT baseline using two-sample z-test.

geometric patterns [Cho19] and 3D spatial information [AMKK21, HZC⁺23]. [FML⁺22] investigated spatial reasoning capabilities of transformer-based models in the UI grounding setting. On the other hand, some works adopted in-context learning, leveraging LLMs for general purpose reasoning to convert spatial information to logic forms [YIL23], or as a general pattern machine for sequence transformation [MXF⁺23]. Recently, several works focused on evaluating spatial reasoning of LLMs as cognitive capability on navigation [YBL⁺23] and planning tasks [MHV⁺24] among various spatial structures. While most existing works rely on linguistic semantics and verbal reasoning, and might not always necessitate spatial awareness, we propose to elicit mind's eye of LLMs in spatial reasoning tasks with various formats from a cognitive perspective. The VoT prompting induces LLMs to create *mental images* for visualizing their internal states and inform subsequent reasoning step.

**World Models of LLMs** While there have been many theoretical debates about whether LLMs can effectively learn an internal world model from ungrounded form alone [BHT⁺20, MGSS21], [LeC22] advocated that world models should represent percepts and action plans at multiple levels of abstraction and multiple time scales, with the capability of planning, predicting, and reasoning. [LWG⁺22] proposed to ground LLM in the physical world by reasoning over the experimental results predicted by external simulation. [HGM⁺23] further leveraged LLMs as world models to predict the subsequent states by action simulation, given predefined states and actions per task. On the other hand, an increasing number of studies focus on investigating internal representations of LLMs. [PP22, AKH⁺21] showed that by utilizing in-context learning, LLMs' learned representations can be mapped to grounded perceptual and conceptual structure in color and spatial domains. Moreover, [GT23] and [NLW23] discovered linear representations of space, time and game state in specifically trained LLMs, which are important for dynamic causal world models. Our work does not probe the internal representations of specialized LLMs, nor does it depend on external simulation engine or state definitions. We demonstrate LLMs' zero-shot capability of representing their percepts at an abstract level, predicting and tracking the internal states over time to generate action plans in multi-hop spatial reasoning tasks, which possibly mirrors the causal world model within LLMs.

# 7 Conclusion

This study introduces Visualization-of-Thought Prompting (VoT), inspired by the human cognitive function of visualizing and manipulating mental images through the mind's eye. We have demonstrated that VoT enables LLMs to exhibit the mechanism of "the mind's eye", as evidenced by their performance in multi-hop spatial reasoning tasks and our comprehensive analysis of the reasoning traces. Remarkably, VoT enable LLMs to outperform state-of-the-art multimodal large language models (MLLMs) in the tested visual tasks. While VoT demonstrates impressive efficacy in LLMs, this emergent capability to create *mental images* to enhance spatial reasoning resembles the mind's eye process, suggesting its promise in MLLMs.

Building on the success of experiments with GPT-4, we plan to investigate how VoT can futher elicit "the mind's eye" in MLLMs to enhance their spatial awareness. Additionally, our future efforts will focus on automatic data augmentation from real-world scenarios, aiming to identify effective methods for learning generalized internal representations of *mental images*. This will further improve the mind's eye of LLMs, ultimately contributing to the advancement of their cognitive and reasoning abilities.

## Limitations

This work only scratches the surface of spatial reasoning of LLMs. Both *mental images* and visual state tracking rely on the emergent ability of advanced LLMs. Therefore, it might cause performance deterioration in less advanced language models or more challenging tasks. Besides, due to the limited data exposure and a lack of explicit instruction tuning, visual state tracking of current LLMs are sensitive to prompts. For example, when explicitly prompted with "use ascii-art", the tracking rate will significantly increase thereby boosting performance, while removing "reasoning" from the **VoT** prompt will cause a decrease of tracking rate. Moreover, the *mental images* tested in our work are limited to 2D grid. To strength the mind's eye of LLMs, more diverse and complicated representation should be explored in the future, such as complex geometric shapes and even 3D semantics shown in Figure 11 in appendix.

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

## A  Synthetic Data

### A.1  Visual Navigation

As depicted in 1, given a specific $k$, the process of generating a 2D navigation map is composed of 3 steps, which are instruction generation, instruction simulation, map rendering. In instruction generation step, we enumerate all possible instruction sets navigating from the starting point to the destination (e.g move up, then move right). During this step, only the direction of each instruction is determined, while the moving distance is undetermined until next step. In instruction simulation step, simulation is applied in the 2D coordinate system with origin (0, 0) as the starting point. To guarantee an unique answer in each navigation map, the moving distance of each instruction is dynamically calculated to avoid overlapping. Each time when an overlapping is detected, the moving distance of previous instruction will be increased by 1 unit recursively until overlapping is resolved. As the distance is determined, those corresponding points are added to the navigating path. After all instructions are completed, the final point is marked as the destination. In the map rendering step, the bounding box of those points is adopted and normalized to a 2D square grid. The starting point and destination are marked with dedicated squares, and cells along the path are marked by empty squares, while other untouched cells are marked by obstacle squares.

---

**Algorithm 1:** Navigation Map Generation

**Input**  : $k$
**Output** : $\mathcal{C}_{solution} = \{s_1, s_2, ..., s_n\}, \mathcal{C}_{textual\_map} = \{t_1, t_2, ..., t_n\},$
$\qquad\quad \mathcal{C}_{visual\_map} = \{v_1, v_2, ..., v_n\};$ **where** $n = 2^{k+1}$

1  $dirs \leftarrow [\text{up}, \text{left}, \text{down}, \text{right}]$
2  $instruction\_sets \leftarrow \text{gen\_instruction}(dirs, k)$
3  **for** $dir\_instructs$ **in** $instruction\_sets$ **do**
4  $\quad$ $cur\_pos, path\_points, moves \leftarrow \text{initialize}()$ $\qquad$ // origin as the starting point
5  $\quad$ **for** $direction$ **in** $dir\_instructs$ **do**
$\qquad\qquad\qquad\qquad\qquad\qquad\qquad\qquad\qquad\qquad\qquad\qquad\qquad\quad$ // instruction simulation
6  $\quad\quad$ **if** *not validate_plan(cur_pos, direction, path_points)* **then**
7  $\quad\quad\quad$ increase_previous_move$(cur\_pos, moves, path\_points)$
8  $\quad\quad$ **end**
9  $\quad\quad$ $cur\_pos, path\_points \leftarrow \text{step\_forward}(cur\_pos, direction, path\_points)$
10  $\quad\quad$ $moves \leftarrow moves \cup direction$
11  $\quad$ **end**
12  $\quad$ $s_i \leftarrow moves$
13  $\quad$ $t_i, v_i \leftarrow \text{render\_map}(\text{extract\_bounding\_box}(path\_points))$
14  $\quad$ $\mathcal{C}_{solution} \leftarrow \mathcal{C}_{solution} \cup s_i$
15  $\quad$ $\mathcal{C}_{textual\_map} \leftarrow \mathcal{C}_{textual\_map} \cup t_i$
16  $\quad$ $\mathcal{C}_{visual\_map} \leftarrow \mathcal{C}_{visual\_map} \cup v_i$
17  **end**

---

Since the direction of each navigation instruction is alternating, there are $4 * 2^{k-1} = 2^{k+1}$ kinds of spatial configurations for a $k$-hop navigation map. During the implementation, we simplify the recursive implementation with an early quit when path overlapping could not be resolved within one iteration, the main consideration of which is the size of the map. So the number of generated map is slightly lower than $2^{k+1}$ as the navigating step $k$ increases.

### A.2  Visual Tiling

The data generation process comprises 3 stages, including configuration generation, question generation and polyomino rendering. In the configuration generation stage, to generate valid spatial configurations of a rectangle and the corresponding polyomino set, we convert a tiling problem to existing formalized problems. One of the problems is an exact cover problem leveraging dancing link algorithm [Knu00], which could be described as: given a matrix of 0s and 1s, find a set of rows containing exactly one 1 in each column. The conversion is to construct a matrix of 0s and 1s, each row of which represents a possible arrangement of placing a specific polyomino in a rectangle. As illustrated in Equation 9, given $k$ polyomino pieces, and a rectangle of $n$ units to be filled, the first $k$

columns compose an one-hot vector indicating the corresponding polyomino, and the last $n$ columns are marked with 0 or 1 depending on whether the corresponding unit is filled by that polyomino. Then finding a set of polyomino arrangements in a rectangle equals to find a set of rows containing exactly one 1 in each column. Another adaptable problem is the boolean satisfiability problem (commonly known as SAT), for which efficient solvers exist [ES03, GN07]. A tiling problem can be converted to SAT by introducing a boolean variable for each possible arrangement of each piece, and then adding clauses comprising of those boolean variables that ensure at least one arrangement of each piece is achieved, while avoiding conflicts between arrangements of one piece or two different pieces.

Given the size of a rectangle and polyominoes to be fit, multiple corresponding solutions are generated by applying those algorithms. Then in the question generation stage, we randomly mask several polyomino pieces in the rectangle, and generate a question answer(QA) pair for each masked polyomino. Finally the rectangle and each polyomino piece are rendered with emoji squares.

$$
\begin{array}{c}
\begin{array}{cc} C_1 \cdots\cdot C_k & C_{k+1} \cdots\cdots\cdots\cdots C_{n+k} \end{array} \\
\begin{array}{c} P.1 \\ \vdots \\ \vdots \\ Pk \end{array}
\left[
\begin{array}{c|cccc}
1 \cdots\cdots 0 & 1 & 0 \cdots 0 & 1 \\
1 \cdots\cdots 0 & 0 & 1 \cdots 1 & 0 \\
\vdots & & & \vdots \\
\vdots & & & \vdots \\
0 \cdots\cdots 1 & 1 & 1 \cdots 0 & 0 \\
0 \cdots\cdots 1 & 0 & 0 \cdots 1 & 1
\end{array}
\right]
\end{array} \tag{9}
$$

### A.3 Visual Data Rendering

After gathering the textual dataset of 2D square grid, we generate the corresponding visual dataset by drawing text onto an image. Specifically we adopt color emojis for a fair comparison as they're more visual friendly to a multimodal model.

### A.4 Dataset Details

Data distribution among various difficulty levels for visual navigation tasks and visual tiling tasks are provided in Table 4 and 5. It provides flexible difficulty control across different tasks. For visual tiling task, the difficulty is controlled by the number of masked polyomino pieces. As the number increases, the more spatial arrangements LLMs need to consider. Regarding the visual navigation task, as illustrated in figure2, we use the number of roads $k$ to control difficulty, which is corresponding to the size of the map.

| Task | K Step | | | | | | Total |
|---|---|---|---|---|---|---|---|
| | **2** | **3** | **4** | **5** | **6** | **7** | |
| **Route Planning** | 8 | 16 | 32 | 64 | 128 | 248 | 496 |
| **Next Step Prediction** | 8 | 32 | 96 | 256 | 640 | 1488 | 2520 |

Table 4: Data distribution of visual navigation dataset with the total navigating step of $k$ indicating difficulty level. The reason why the number of generated map is slightly lower than $2^{k+1}$ for $k > 5$ is explained in Appendix A.1.

| | Mask count | | Total |
|---|---|---|---|
| | **2** | **3** | |
| **Configuration** | 248 | 124 | 376 |
| **QA Instance** | 489 | 307 | 796 |

Table 5: Details of visual tiling dataset. Some QA instances are discarded when multiple solutions exist and all answers are correct.

## B Examples

For visual navigation and visual tiling tasks, the structured input template is comprised of task instruction, input parameters and prompt of specific setting.

### B.1 Visual Tasks

Task instructions and responses of each visual task under setting GPT-4 VoT are provided as following:

- Route Planning Task instruction in Figure 6, response in Figure 12 .

- Next Step Prediction Task instruction in Figure 6, response in Figure 13.
- Visual Tiling Task instruction in Figure 7, response in Figure 14.

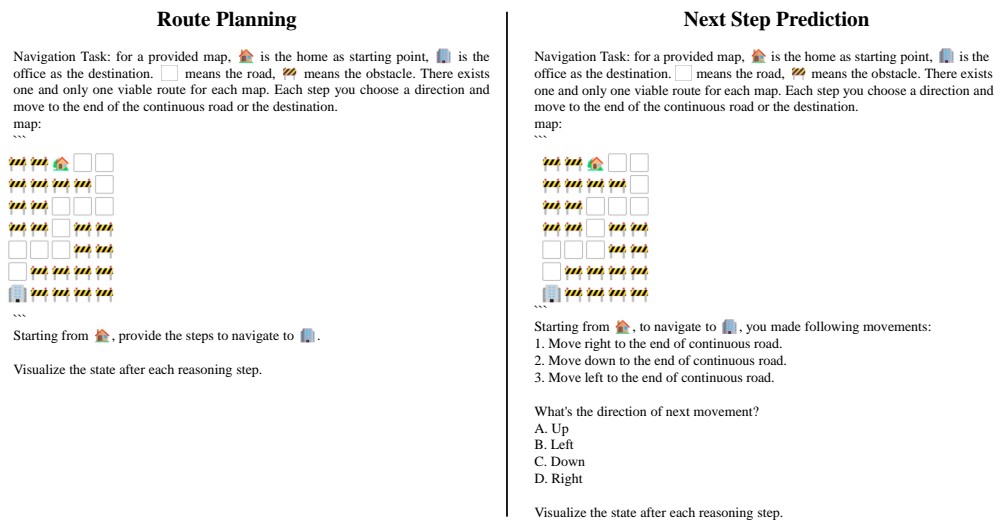

Figure 6: Task Instruction of visual navigation.

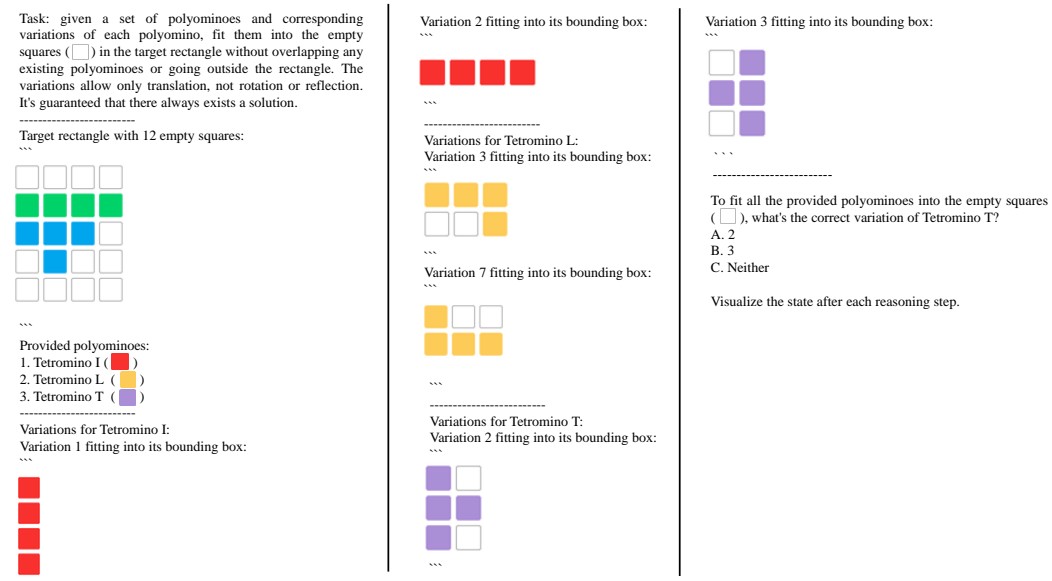

Figure 7: Task Instruction of visual tiling.

## B.2 Natural Language Navigation

**Prompt Example**   You have been given a 3 by 3 square grid. Starting from a vertex, you will move along the edges of the grid. Initially, you are positioned at the bottom-left corner of the grid, where you will find a torch, then you go right, where you will find an infant bed, then you go right, where you will find an American dipper. Then you go up, where you will find a jay, then you go left, where you will find a terrapin, then you go left, where you will find a microwave oven. Then you go up,

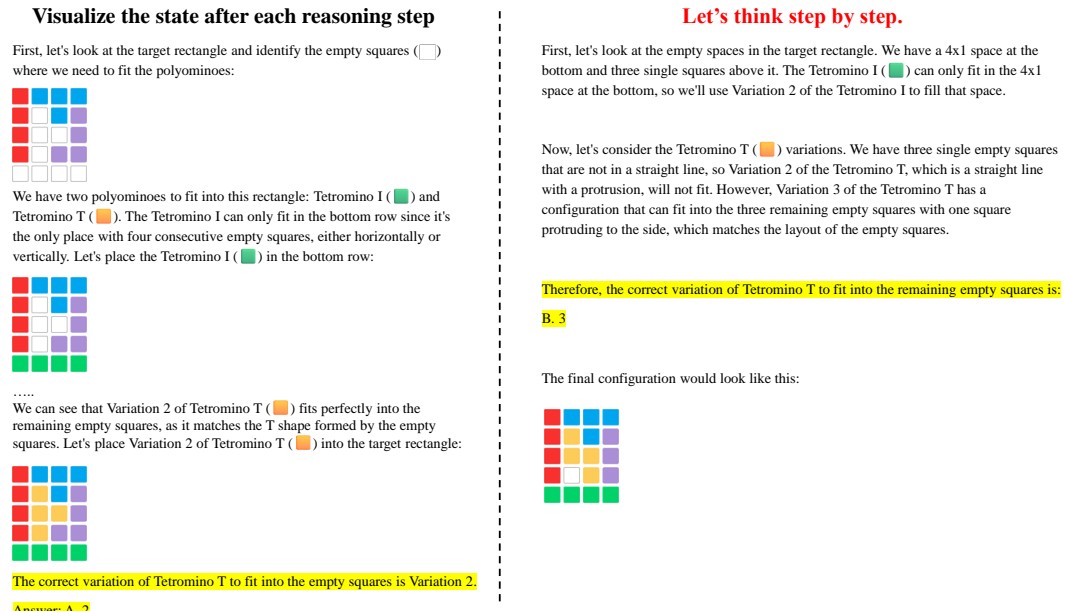

(a) Common behaviors of VoT and CoT prompting in visual tiling task, with the overall track rate of 87.1% and 57.4% respectively.

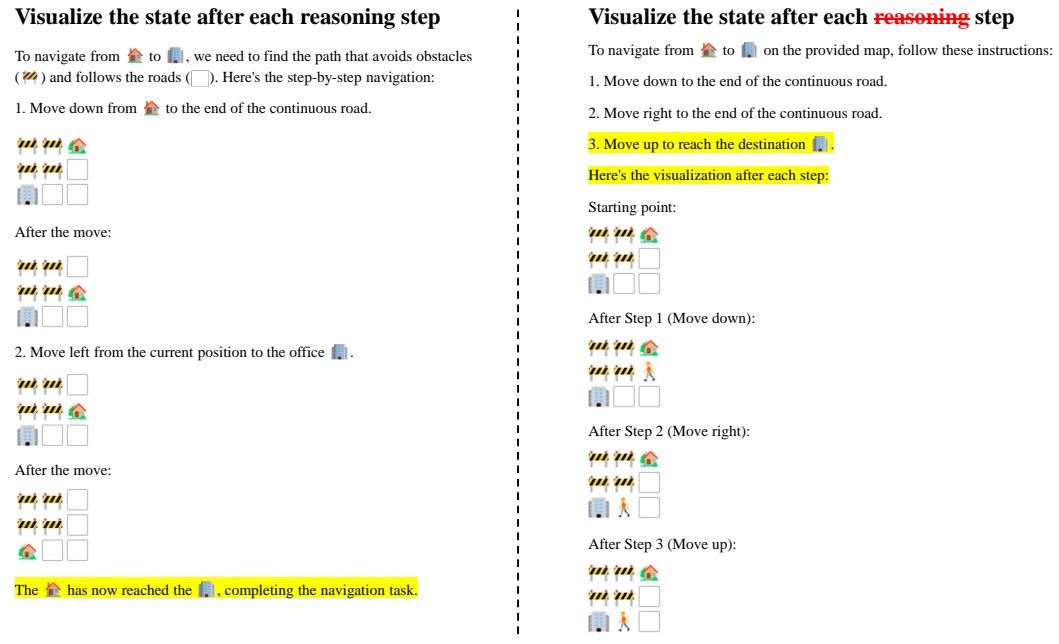

(b) Route planning task is sensitive to prompts. After deleting the word "reasoning" from the VoT prompt, final answer and state is derived without conditioning on state visualization in many cases. The left example provides a correct solution with imperfect intermediate visualization, while the right example provides a wrong solution.

Figure 8: Example cases of visual state tracking behaviors in different tasks. We provide side-by-side comparisons to show how visual state tracking is influenced by prompt, where bold texts on the top are prompts to compare. Case on the left exhibits visual state tracking in an interleaved manner, where next state is conditioned on visualization of previous states. Case on the right disentangles verbal reasoning and state visualization, which leads to a drop in tracking rate and task performance.

where you will find a baseball player, then you go right, where you will find a harvestman, then you go right, where you will find a neck brace. Now you have all the information on the map. You start at the position where the torch is located, then you go right by one step, then you go right by one step, then you go up by one step, then you go up by one step, then you go left by one step, then you go down by one step, and then you go down by one step. What will you find?

**Response Example**    See Figure 15.

## C    Visual State Tracking

As for where this emergent ability stems from, it might derive from tabular data, city grid navigation, maze exploration related coding problems [YBL+23]. These tasks involves understanding and manipulating objects in a 2D square grid. Besides, we conjecture the exposure of ascii-art comments [Reg19] during LLMs' code pre-training possibly enhances this generalized ability. As a fact to support this conjecture, the visual tiling task is different from navigation tasks because it requires shape understanding and spatial manipulation ability. While tabular data and square grid navigation data boost row-wise or column-wise attention, ascii-art supplements intricate spatial attention to understand and manipulate 2D shapes. Additionally, ascii-art in code comments is presented in various formats, one of which is interleaved ascii diagrams, natural language and programming language. It require LLMs to generate the interleaved *mental images* and text sequence, thereby enhancing spatial visualization ability and spatiotemporal causality. Interestingly in the natural language navigation task, when GPT-4 is prompted with "use ascii-art to visualize", the complete tracking rate increases to 98.5% (+78.5%), boosting task performance to 62.5% (+3.5%).

### C.1    Ascii-art in Code Comments

Ascii-art is commonly used in code comments to represent data structure, diagram, geometry and so on, which could benefit LLMs' spatial understanding and visualization capability. Besides, it's also used to illustrate how an algorithm works or simulate an operation, where reasoning traces and corresponding visualization are presented in an interleaved manner. Below are several examples in open-source projects.

- **Spatial Causality**:Double-ended queue in Rust, Scrolling web pages and tree rotation present triplets of previous visual state, instruction, and updated state of instruction following.
- **Temporal Causality**: Undo systems from emacs provides various temporal states of the undo system when undo operation happens in different timelines and corresponding visualizations in an interleaved manner. Each visualization reflects the temporal casuality of the system state.

This kind of interleaved sequence tracks the system state over time, thus reflecting spatiotemporal casuality.

## D    Performance Trends Across Levels

In this analysis, we examine performance trends across varying difficulty levels in the next-step prediction task for models utilizing either CoT or VoT methods. These trends are crucial for understanding the inherent unpredictability associated with random guessing. As $k$ increases from 2 to 7 in a $k$-step navigation map, distinct performance patterns emerge among different models, as depicted in Figure 9. Larger language models such as GPT-4 and LLAMA3-70B demonstrate a more predictable decrease in accuracy with increasing $k$. This trend indicates a robust ability to handle progressively challenging tasks, despite the overall decrease in performance. Detailed statistics are provided in table 6. In contrast, less powerful models like GPT-3.5 and LLAMA3-8B exhibit an irregular performance trajectory. These models show variable accuracy, with significant fluctuations at higher difficulty levels, suggesting a reliance on random guessing, particularly under conditions of increased task difficulty. This behavior highlights their limitations in sustaining reliable reasoning processes through more complex scenarios. Furthermore, the VoT method seems to offer a modest improvement in performance for the less powerful models, particularly in scenarios of lower difficulty.

This observation suggests that VoT might be advantageous for enhancing reliable reasoning in simpler spatial reasoning tasks, potentially compensating for the inherent weaknesses of smaller language models.

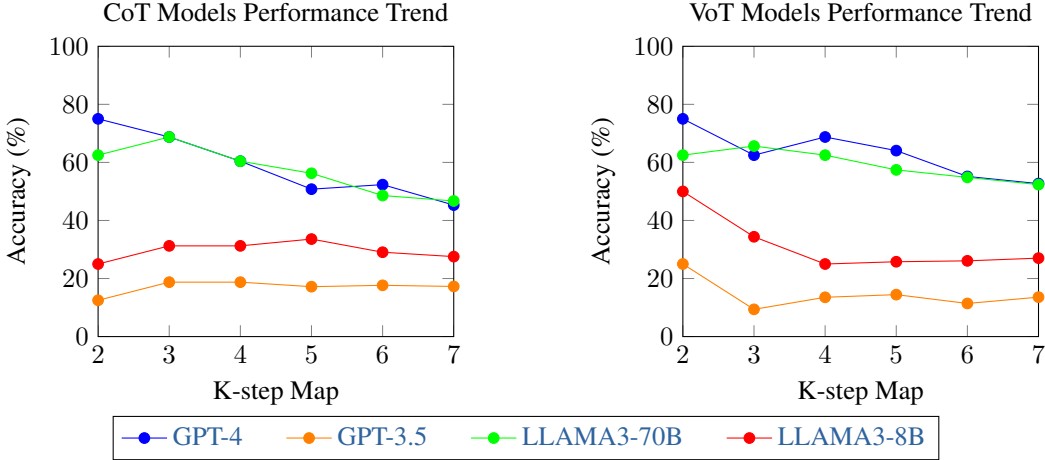

Figure 9: Performance Trends of CoT and VoT Models Across difficulty levels in next-step-prediction task.

| Model | K-step Map | Map Count | CoT Accuracy (%) | VoT Accuracy (%) |
|---|---|---|---|---|
| GPT-4 | 2 | 8 | 75.00 | 75.00 |
| | 3 | 32 | 68.75 | 62.50 |
| | 4 | 96 | 60.42 | **68.75** |
| | 5 | 256 | 50.78 | **64.06** |
| | 6 | 640 | 52.34 | **55.16** |
| | 7 | 1488 | 45.30 | **52.69** |
| LLama3-70B | 2 | 8 | 62.50 | 62.50 |
| | 3 | 32 | 68.75 | 65.63 |
| | 4 | 96 | 60.42 | **62.50** |
| | 5 | 256 | 56.25 | **57.42** |
| | 6 | 640 | 48.59 | **54.84** |
| | 7 | 1488 | 46.71 | **52.35** |

Table 6: CoT and VoT performance of advanced models in next-step-prediction task across various difficulty levels. While performance drops as difficulty level increases, VoT method generally maintains a higher accuracy compared to CoT, highlighting its robustness in more challenging scenarios.

# E   Case study

We consider visual state tracking similar to spatiotemporal simulation. During the simulation in those tasks, we discovered several interesting behaviors of LLM.

**1.** Diverse visualization formats for state tracking: Nearly 30 different symbols found in the navigation tasks to track the navigation progress, including marking the path, marking the current location. Among those diverse representations, LLM succeeded in some challenging cases where it used directional arrow emojis to indicate both the location and moving direction at each step. More examples could be found in Appendix E.1.

**2.** Inconsistency between language and visualization: This is commonly observed across all tasks. Due to the limited visualization capability, sometimes LLM generates accurate language instruction but inaccurate visualization. And in other cases, LLM generates wrong answers even the visualization is generated correctly, which reflects its limitation of spatial understanding as discussed in previous section. More examples could be found in Appendix E.2.

**3.** Self-refine mechanism: We found several cases in visual tiling tasks where spatial hallucination happens due to the inconsistency or inaccurate visualization. Subsequently, LLM refined its reasoning, resulting in an accurate visualization and the correction of the final answer. More examples could be found in Appendix E.3.

### E.1 *mental images* for State Tracking

In the visual navigation task, LLM adopted various symbols and representations to track the state of navigation progress. As shown in Figure 10, there're several tracking styles.

- Mark the path: adopting an identical symbol to mark current location or part of the path.
- Mark path and direction: using directional arrows to mark current location and indicate the moving direction simultaneously, which is more challenging than simply marking the path.
- Mark path with temporal steps: using numbers to demonstrate both temporal steps and current location.
- Remove road: turning roads into obstacles to avoid turning back, instead of adopting additional symbols to mark the path.

### E.2 Inconsistency between Language and Visualization

In the visual tiling task, two inconsistent steps are highlighted in Figure 16. One is the inconsistent visualization with the language instruction of "place Variation 6 of Tetromino L". Another is the wrong decision to chose "Variation 2 of Tetromino I" given the visualization of the valid state.

### E.3 Self-refine Mechanism

We found visualization could enhance LLM's reasoning by self-grounding and refining subsequent reasoning steps in some cases. As shown in Figure 17, despite successfully identifying variation 1 of tetromino L as incorrect option, GPT-4 excluded the correct option of variation 6 even it's placed accurately due to spatial hallucination (overlapping with yellow pieces), which led to a impossible solution. Then it detected the mistake and re-evaluate the placement of variation 6. Finally it placed the correct piece into the top left corner and validated the answer by filling the remaining space.

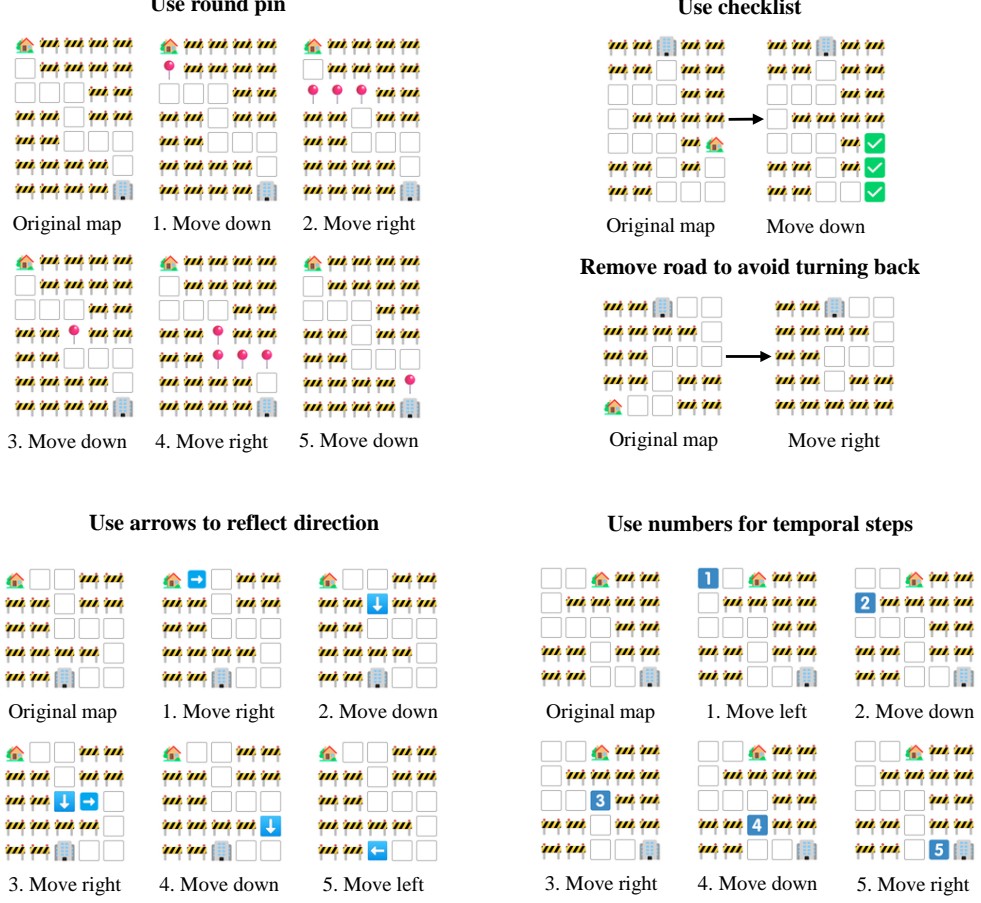

Figure 10: *Mental images* for state tracking in visual navigation tasks. We found diverse visualization formats in LLMs to track the navigation progress, including marking the path, marking the current location, marking directional steps, and marking temporal steps. More details are available in Appendix E.1

The ASCII art you've provided represents a three-dimensional cube with labeled vertices from 0 to 7. To unfold the front face (which appears to be the face connecting vertices 1, 2, 6, and 5), we would lay it flat. Assuming that the front face is the one with vertices 1, 2, 6, and 5, and we unfold it downwards, the net might look something like this:

```
   0     3
    *------*
   /|      /|
  *------* |
 4||     ||7
  | *-----|-*
  |/      |/
  *------*
  1     2
```

Now, let's unfold the front face (1, 2, 6, 5) downwards:

```
   0     3
    *------*
   /      /|
  *------* |
 4|      ||7
  |      |*
  |      |/
  *------*
  |      |
  |      |
  *------*
  1     2
```

Here, the face with vertices 1, 2, 6, and 5 has been unfolded downwards. The vertical lines connecting the top face to the front face have now become horizontal, indicating that the front face is lying flat. The cube is now represented as a two-dimensional net, with the front face unfolded. Keep in mind that this is a simplified representation and that in an actual net, the faces would be connected at the edges without the gaps shown in the ASCII art.

Ascii cube:

```
   0     3
    *------*
  1/|   2/|
   *------* |
   | |4  | |7
   | *-----|-*
   |/     |/
   *------*
   5     6
```

Unfold the front face

Figure 11: Representation for 3D semantics and qualitative results of GPT-4 manipulating the *mental image* correctly.

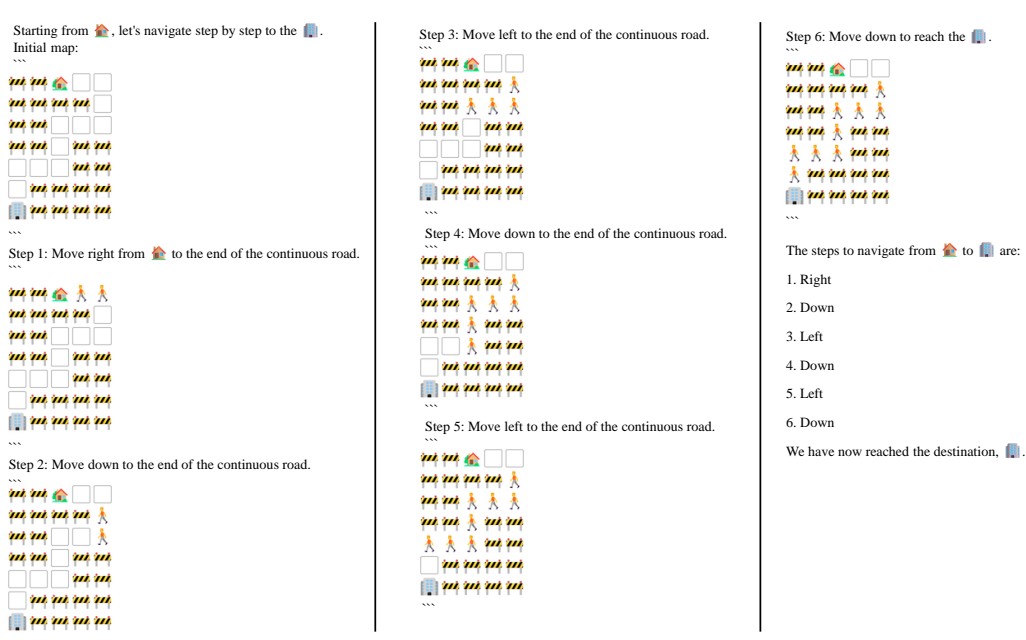

Figure 12: Responses of Route Planning

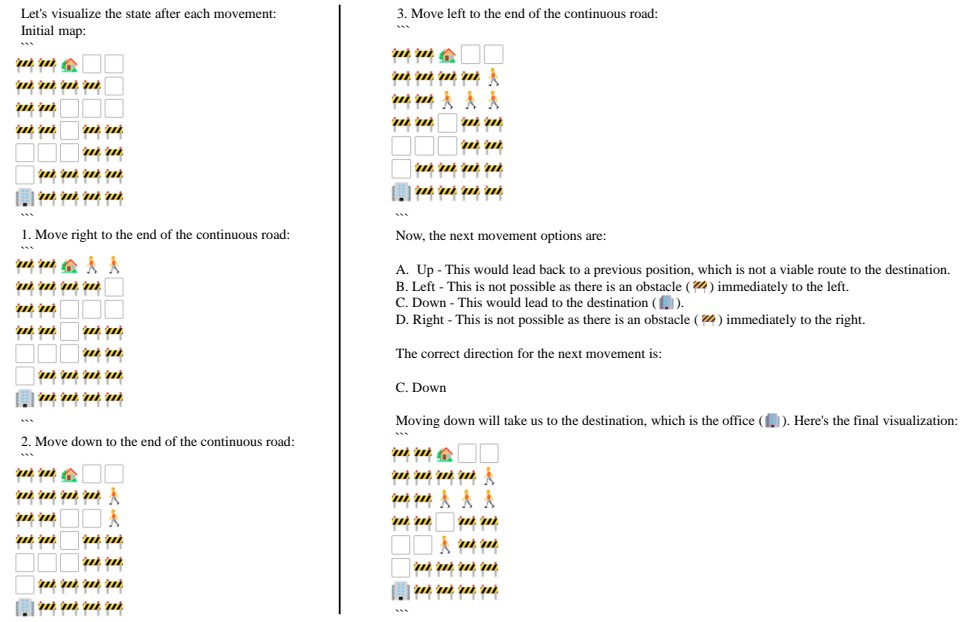

Figure 13: Response of Next Step Prediction

Let's start by examining the empty spaces in the target rectangle and the provided polyominoes and their variations. We need to fit the Tetromino I, Tetromino L, and Tetromino T into the empty spaces without overlapping any existing polyominoes or going outside the rectangle.

The target rectangle with 12 empty squares looks like this:

```

```

We have two variations for Tetromino I ( 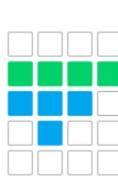 ), which can either be placed vertically or horizontally. Given the configuration of the empty spaces, the only way to place the Tetromino I is horizontally at the top row, as the vertical placement would not fit anywhere.

After placing Tetromino I ( 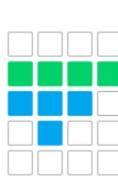 ), the target rectangle looks like this:

```

```

Next, we have two variations for Tetromino L ( 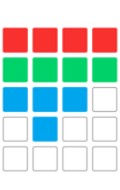 ). We can see that Variation 3 of Tetromino L can fit in the bottom left corner, while Variation 7 would not fit anywhere without overlapping or going outside the rectangle.

After placing Tetromino L ( 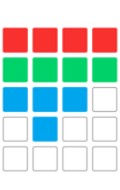 ), the target rectangle looks like this:

```

(a) Response of Visual Tiling - Page 1

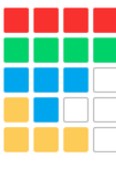

```

Now, we have two variations for Tetromino T ( ). We need to fit this into the remaining empty spaces. Variation 2 of Tetromino T cannot fit because it would require two empty spaces next to each other vertically, which we do not have. However, Variation 3 of Tetromino T fits perfectly into the remaining space.

After placing Tetromino T ( ), the target rectangle looks like this:

```

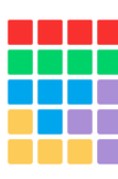

```

The correct variation of Tetromino T that fits into the empty squares is:

B. 3

(b) Response of Visual Tiling - Page 2

Figure 14: Responses of Visual Tiling

Let's visualize the grid with the items you've described:

```
+---+---+---+
| S | B | M |
+---+---+---+
| P | C | V |
+---+---+---+
| T | W | C |
+---+---+---+
```

Where:

- T = Cassette player
- W = Wool
- C = Conch
- V = Moving van
- C = Confectionery store
- P = Pot pie
- S = Siamang
- B = Black-and-white colobus
- M = Minivan

You start at the bottom-left corner where the cassette player is located (T). Now, let's follow your steps:

1. Go right by one step (to Wool - W):

```
+---+---+---+
| S | B | M |
+---+---+---+
| P | C | V |
+---+---+---+
| T |*W*| C |
+---+---+---+
```

2. Go right by one step (to Conch - C):

```
+---+---+---+
| S | B | M |
+---+---+---+
| P | C | V |
+---+---+---+
```

(a) Response of Natural Language Navigation - Page 1

```
| T | W |*C*|
+---+---+---+
```

3. Go up by one step (to Moving van - V):

```
+---+---+---+
| S | B | M |
+---+---+---+
| P | C |*V*|
+---+---+---+
| T | W | C |
+---+---+---+
```

4. Go left by one step (to Confectionery store - C):

```
+---+---+---+
| S | B | M |
+---+---+---+
| P |*C*| V |
+---+---+---+
| T | W | C |
+---+---+---+
```

5. Go left by one step (to Pot pie - P):

```
+---+---+---+
| S | B | M |
+---+---+---+
|*P*| C | V |
+---+---+---+
| T | W | C |
+---+---+---+
```

6. Go up by one step (to Siamang - S):

```
+---+---+---+
|*S*| B | M |
+---+---+---+
| P | C | V |
+---+---+---+
| T | W | C |
+---+---+---+
```

7. Go right by one step (to Black-and-white colobus - B):

(b) Response of Natural Language Navigation - Page 2

Figure 15: Responses of Natural Language Navigation

```
+---+---+---+
| S |*B*| M |
+---+---+---+
| P | C | V |
+---+---+---+
| T | W | C |
+---+---+---+
```

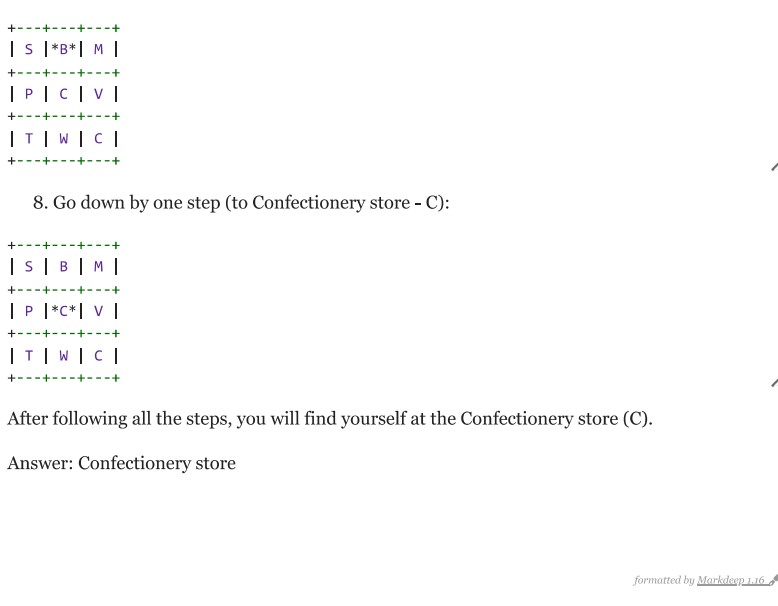

8. Go down by one step (to Confectionery store – C):

```
+---+---+---+
| S | B | M |
+---+---+---+
| P |*C*| V |
+---+---+---+
| T | W | C |
+---+---+---+
```

After following all the steps, you will find yourself at the Confectionery store (C).

Answer: Confectionery store

(c) Response of Natural Language Navigation - Page 3

Figure 15: Responses of Natural Language Navigation

First, let's look at the target rectangle and identify the empty squares ( ▢ ):

```

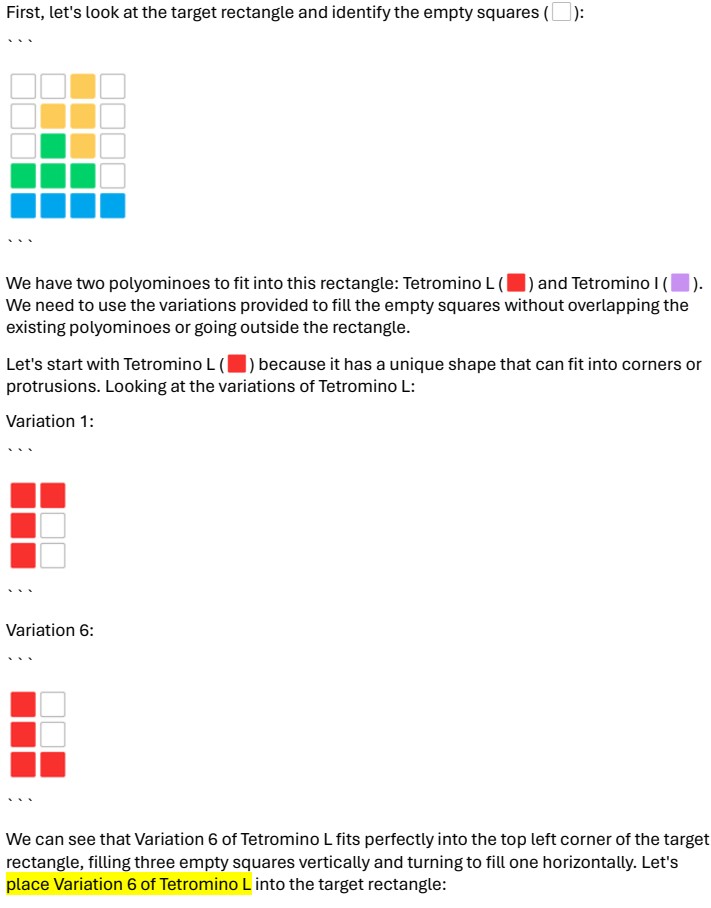

```

We have two polyominoes to fit into this rectangle: Tetromino L ( 🟥 ) and Tetromino I ( 🟪 ). We need to use the variations provided to fill the empty squares without overlapping the existing polyominoes or going outside the rectangle.

Let's start with Tetromino L ( 🟥 ) because it has a unique shape that can fit into corners or protrusions. Looking at the variations of Tetromino L:

Variation 1:

```

Variation 6:

```

We can see that Variation 6 of Tetromino L fits perfectly into the top left corner of the target rectangle, filling three empty squares vertically and turning to fill one horizontally. Let's place Variation 6 of Tetromino L into the target rectangle:

```

(a) Inconsistency between language and visualization - Page 1

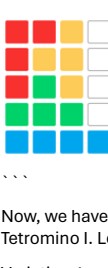

```

Now, we have four empty squares remaining in a straight line, which matches the shape of Tetromino I. Let's look at the variations of Tetromino I:

Variation 1:

```
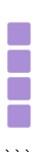
```

Variation 2:

```
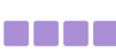
```

Since the remaining empty squares are in a horizontal line, ==Variation 2 of Tetromino I will fit perfectly==. Let's place Variation 2 of Tetromino I into the target rectangle:

```
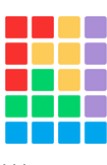
```

The correct variation of Tetromino I that fits into the target rectangle is:

B. 2

(b) Inconsistency between language and visualization - Page 2

Figure 16: Inconsistency between language and visualization

Let's start by visualizing the target rectangle with the empty squares ( 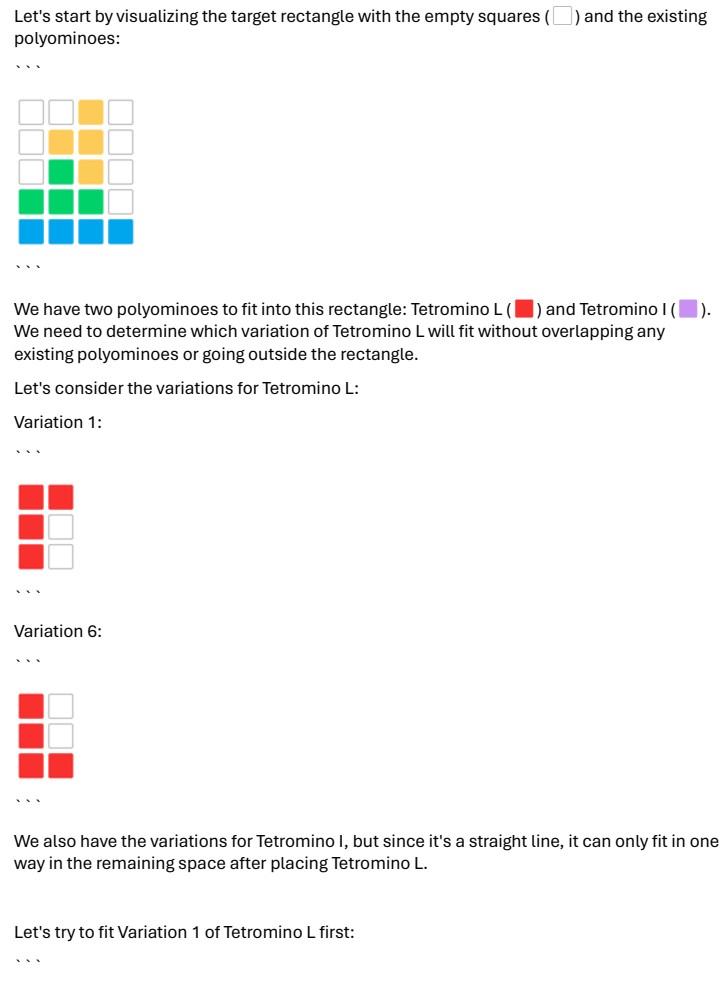 ) and the existing polyominoes:

```
```

We have two polyominoes to fit into this rectangle: Tetromino L (  ) and Tetromino I (  ).
We need to determine which variation of Tetromino L will fit without overlapping any existing polyominoes or going outside the rectangle.

Let's consider the variations for Tetromino L:

Variation 1:

```
```

Variation 6:

```
```

We also have the variations for Tetromino I, but since it's a straight line, it can only fit in one way in the remaining space after placing Tetromino L.

Let's try to fit Variation 1 of Tetromino L first:

```

(a) Example case of Self-refine - Page 1

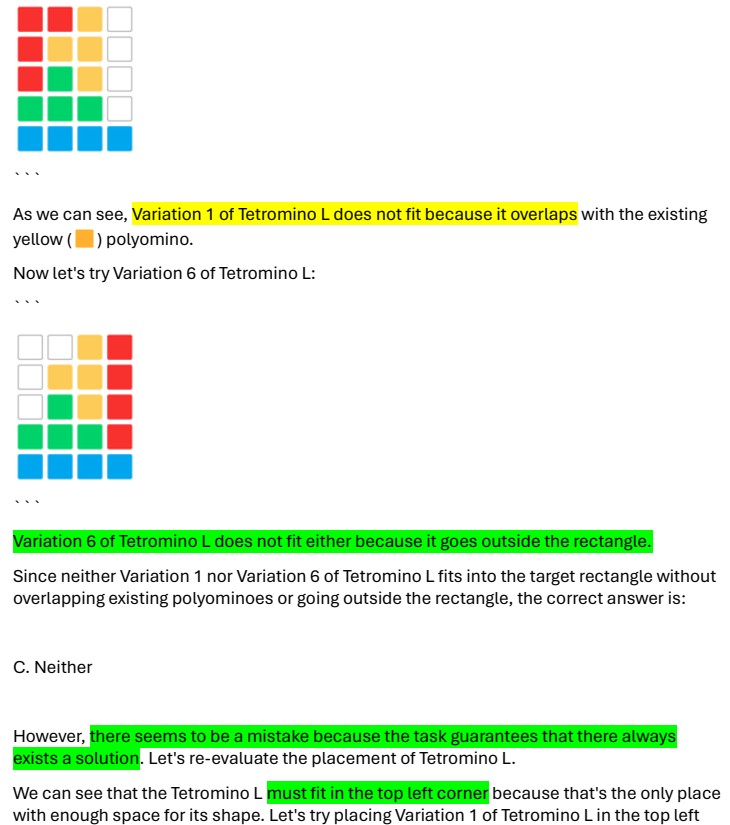

```

As we can see, Variation 1 of Tetromino L does not fit because it overlaps with the existing yellow ( ◻ ) polyomino.

Now let's try Variation 6 of Tetromino L:

```

```

Variation 6 of Tetromino L does not fit either because it goes outside the rectangle.

Since neither Variation 1 nor Variation 6 of Tetromino L fits into the target rectangle without overlapping existing polyominoes or going outside the rectangle, the correct answer is:

C. Neither

However, there seems to be a mistake because the task guarantees that there always exists a solution. Let's re-evaluate the placement of Tetromino L.

We can see that the Tetromino L must fit in the top left corner because that's the only place with enough space for its shape. Let's try placing Variation 1 of Tetromino L in the top left corner:

```

(b) Example case of Self-refine - Page 2

Figure 17: Example case of Self-refine

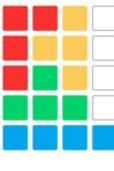

```

This time, Variation 1 of Tetromino L fits perfectly in the top left corner. Now, we can place the Tetromino I in the remaining space:

```

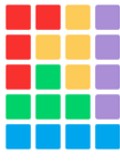

```

The Tetromino I fits perfectly in the remaining space using Variation 1 (vertical placement). Therefore, the correct variation of Tetromino L that fits into the target rectangle is:

A. 1

The initial attempt to place Variation 1 of Tetromino L was incorrect because it was not placed in the top left corner. Upon correcting the placement, we find that Variation 1 does indeed fit, and the Tetromino I can be placed afterward to complete the puzzle.

(c) Example case of Self-refine - Page 3

Figure 17: Example case of Self-refine

