# OpenReview forum: "Mind's Eye of LLMs: Visualization-of-Thought Elicits Spatial Reasoning in Large Language Models"
_NeurIPS.cc/2024/Conference — NeurIPS 2024 poster_

### Official Review · Reviewer_Af54 · 2024-07-05

**Soundness:** 4
**Presentation:** 4
**Contribution:** 4
**Rating:** 8
**Confidence:** 4

**Summary:**

The paper explores the spatial reasoning capabilities of LLMs. To this end, the paper introduces novel testbeds including multiple tasks and, importantly, a novel approach called Visualization-of-Thought (VoT). This method aims to enhance the spatial reasoning capabilities of LLMs by visualizing their reasoning processes, also inspired by chain of thought CoT reasoning and human cognitive abilities.
Evaluating on the mentioned testbed the authors show that VoT significantly improves the spatial reasoning performance of LLMs, outperforming e.g. CoT reasoning.

**Strengths:**

- The introduction of Visualization-of-Thought (VoT) represents a novel problem and significant advancement in eliciting spatial reasoning in LLMs, a relatively unexplored area.
- The paper is well-structured, balancing necessary details in the main text with extensive supplementary information in the appendix. The problem, tasks as well as the VoT approach are well motivated and described.
- The paper presents well-conducted and detailed experiments across multiple tasks, providing robust evidence of VoT’s effectiveness. Further, the experimental setup is well described. Additionally, the consideration of openly available models (LLaMA), enhances the study’s reproducibility.
- The authors provide a thorough discussion of the results and the limitations providing valuable insights for future work.

**Weaknesses:**

While novel, the introduced and investigated tasks to evaluate spatial reasoning capabilities of LLMs may still be relatively simple compared to real-world spatial reasoning challenges. However, this is negligible. The paper’s contribution provides significant novelty and lays a robust foundation for future exploration and advances of spatial reasoning in large language models.

**Questions:**

- How might VoT be adapted or scaled to work with 3D spatial reasoning tasks or other modalities?

**Limitations:**

The authors have adequately addressed the limitations of their work.

---

> ### Author Rebuttal · Authors · 2024-08-06
>
> Thank you for your feedback, and we appreciate your support!
>
> Spatial reasoning in LLMs is a less explored research topic, we only scratch the surface of it. Despite this testbed is relatively simple to humans and real-world spatial reasoning challenges, it's still challenging for LLMs. It covers various aspects of spatial reasoning, and offers various difficulty levels to appropriately evaluate LLMs.
> > **Question: 3D spatial reasoning tasks and other modalities**
>
>  As LLMs' inherent ability to generate text-form visualizations develops, VoT can be applied to more and more tasks, including 3D tasks. For example, folding a net into a 3D cube, and identifying different perspectives of a 3D cube.
>  Moreover, making VoT work with multi-modal LLMs is one of our primary future works. Besides text-form visualizations, VoT can be adapted to generate visual modality by code generation (indirect) or image generation (direct). The code generation approach might be applicable for proprietary advanced MLLMs at current stage.
>  We'll explore the approach to directly generate images as visualization to guide subsequent reasoning. For open-source MLLMs, they can be adapted to multimodal generation to gain the visualization ability. This will also benefit proprietary MLLMs for two reason. First, they can leverage external visualizers to ground their reasoning and build more powerful applications. Second, the generated multimodal reasoning trajectory is appropriate for data augmentation, which will benefit interested researchers in the community.

---

> > ### Comment · Reviewer_Af54 · 2024-08-12
> >
> > Thank you for the rebuttal. I have no further questions.
> >
> > After carefully considering the other reviews and the authors’ responses, I would keep my initial score.

---

### Official Review · Reviewer_yVwi · 2024-07-08

**Soundness:** 3
**Presentation:** 2
**Contribution:** 3
**Rating:** 5
**Confidence:** 3

**Summary:**

The paper proposes a new prompting method, "Visualization of Thought" (VoT) prompting, to enable LLMs to perform better on spatial reasoning tasks.  In VoT, the LLM is prompted with instructions on performing a multi-step spatial reasoning task, followed by the text "Visualize the state after each reasoning step".  The authors evaluate this prompting method on three types of spatial reasoning tasks, and show that it substantially improves performance on these tasks as compared with Chain-of-Thought prompting, as well as with other variations on prompting.

**Strengths:**

VoT prompting is an interesting and (I think) novel approach to elicit better spatial reasoning in LLMs, and may be useful in multimodal systems as well (this is future work).  The evaluated tasks are chosen to capture important aspects of spatial reasoning abilities. The paper reports on systematic experiments that evaluate these abilities on the chosen tasks.

**Weaknesses:**

There are three categories of weaknesses which I believe can be improved on by the authors.

1. Some aspects of the paper are unclear and need more explanation.

-- The paper says "This method augments LLMs with a visuospatial sketchpad".  It's unclear what is meant here by visuospatial sketchpad.  This sentence makes it sound like there is a new architectural feature added to the LLM, but my understanding from the rest of the paper is that VoT just asks the LLM to generate a series of "visualizations" in its output.  Please make it clearer in the description what exactly you mean.

-- Figure 4 gives examples of VoT prompting.  But it wasn't clear to me whether these prompts include actual images (e.g., the pictures of barriers and people in the navigation task, or the pictures of colored and empty boxes in the visual tiling task) or whether these images in the figures were actually translated into some kind of text encoding for the LLM.  Please make clear exactly what is in the prompt---images or text---and if text, what exactly is the text encoding of the pictures you are showing?

-- Equation 5 defines v_i as "visualizations".  What exactly are these, in terms of LLMs?  Are they text or image outputs from the LLMs?

-- In several figures (e.g., Figure 4, 6, 7) prompts and responses are shown but it needs to be made very clear which part is the prompt and which part the response from the LLM.  These were not clear enough for me.

-- In Figure 8, again, what is part of the prompt and what is part of the response?

-- Figure 10: The caption only says "mental images for state tracking in visual navigation tasks."  Please say more in the caption about what this figure is showing.  Was "Use round pin" part of the prompt, or is this just describing what the LLM did?  What are you trying to show in this figure?  In general, please make the figure captions more informative.

-- "Specifically, we adopt GPT-4 and GPT-4 Vision" -- there are several versions of each of these LLMs on the Azure platform -- please specify which ones you used for these experiments.

-- "We adopted sub-string matching as f_correct to determine correctness."
This didn't really explain how you determined correctness.  Please give more details (enough for someone else to replicate) and explain why this sub-string matching method is reliable for assessing correctness.

2. Scaling:  The tasks used were relatively simple: 3x3 maps for navigation, 5x4 rectangle for tiling.  Did you do any experiments with scaling up the size of these?  How does this method's spatial reasoning abilities scale as the size is increased?

3. Anthropomorphism:  The paper is filled with anthropomorphic language, which makes it difficult to understand.  For example:

-- "LLMs possess the ability to create and manipulate mental images in the mind's eye."   Using terms like "mental images" and "the mind's eye" assumes that the LLM is doing something analogous to the brain, which is not necessarily true!

-- "visual state tracking is implemented by generating the visualization v_i as
a mental image of the internal state si after each reasoning step z".   A mental image is something a human has.  Using that term for LLMs is unhelpful anthropomorphizing.

-- The paper calls the intermediate outputs of LLMs "thoughts" (e.g., "this process by adding a visuospatial sketchpad to each thought z_i") -- again, an unhelpful and confusing anthropomorphism.

-- "We demonstrate LLMs’ zero-shot capability of visualizing their precepts at abstract level". Visualizing is a cognitive ability of brains.  Saying that LLMs "visualize" is an incorrect anthropomorphism.

-- ""this emergent capability to create mental images to enhance spatial reasoning resembles the mind’s eye process, suggesting its promise in MLLMs."  Same comments as above.

**Questions:**

-- I was confused by Figure 8b.  The left side shows the house moving -- is that correct?  Also, it doesn't do what the instructions say: e.g., "move down to the end of the continuous road" -- after the move the house only moved down one square, not two.  Is this considered a correct solution?    The right side shows a person moving.  Where did the notion of moving a person versus moving a house come from?

**Limitations:**

The authors have adequately addressed the limitations.

---

> ### Author Rebuttal · Authors · 2024-08-06
>
> We sincerely thank you for your careful and constructive review of this paper. Your insights are invaluable, and we have carefully considered each of your comments.
> > **Weakness 1: Unclear details**
>
> We'll improve clarity of *visuospatial sketchpad*, *caption of figure 4 and 10* in the camera-ready version. The data construction and evaluation code will be released after paper acceptance to facilitate replication. Below are explanations per question:
> 1. We recognize that "augment" here is misleading. The concept "visuospatial sketchpad" is borrowed from the "Working Memory Model"[1]. We use it to refer to LLMs' ability of retaining visuospatial information within its context, rather than an external tool added to LLMs.
> 2. All the input for LLMs are text only, the map/rectangle input in the tasks is 2D grid comprising of special characters (**line 48**). For example, 🏠, 🚧 and 🟦 are emoji icons in the input that LLMs can understand, and 🚶is generated by LLMs in the output. For GPT-4V, we use Pilmoji library to draw special characters into an image. We discussed this implementation in **appendix A**.
> 3. Visualizations are also text-form (**line 130**).
> 4. We apologize for any confusion caused by Figure 4. It's indended to provides the overview of three tasks without covering all details. It illustrates the input format, VoT prompt and the visualizations generated by LLMs. We'll add descriptions to the caption of Figure 4. Regarding comprehensive details about inputs and responses, they are listed in appendix B.1.
> 5. Figure 8 provides side-by-side comparisons to show the how response is influenced by prompt, where bold texts on the top are prompts to compare.
> 6. We regret the brevity of Figure 10's caption. Figure 10 is to illustrate diverse visualization formats in LLMs to track the navigation progress, including marking the path, marking the current location, and directional/temporal steps. We will include these descriptions in the figure caption and note that detailed descriptions are available in **Appendix E.1**.
> 7. As specified in **section 4.1 (line 152)**, model version of GPT-4 is 1106-preview, and GPT-4V is vision-preview.
> 8. In the natural language navigation task, LLMs often output additional words in the extracted answer besides the expected object name. For example, "Answer: You will find ...". We adopted the substring matching method as previous work did. For visual navigation and visual tiling task, we adopted exact matching. Since sub-string matching can encompass exact matching, we refer to f<sub>correct</sub> implementation using this term for simplicity.
>
> > **Weakness 2: Scaling**
>
> We appreciate the insightful question regarding the scalability of the tasks. Our testbed indeed provide flexible difficulty control across different tasks. For visual tiling task, the difficulty is controlled by the number of empty squares. As the number increases, the more spatial arrangements LLMs need to consider. Regarding the visual navigation task, as illustrated in figure 2, we use the number of roads to control difficulty, which is directly correlated to map size. Data distribution is provided in table 4 and 5.
> Specifically, in the next-step-prediction task, the difficulty is also controlled by the number of navigation instructions. For example, a 7 * 7 map with 6 instructions is significantly more complex than a 3 * 3 map with 1 instruction. The distribution of map size is as follows:
>
> | Height \ Width | 3   | 5   | 7   | 9   |
> |----------------|-----|-----|-----|-----|
> | 3 | 24  | 60  | 116 | 48  |
> | 5 | 60  | 272 | 652 | 120 |
> | 7 | 116 | 652 | 184 | 24  |
> | 9 | 48  | 120 | 24  | N/A    |
>
> The performance of LLMs across these varying difficulty levels is shown in Figure 9. We observe that for advanced LLMs, VoT is more robust to various difficulties compared to CoT. To clearly illustrate the performance gap between VoT and CoT in advanced LLMs across varying difficulty levels, we provide the following comparative table:
>
> | Model       | K-step Map | Map Count | CoT Accuracy (%) | VoT Accuracy (%) |
> |-------------|------------|-----------|------------------|------------------|
> | **GPT-4**   | 2 | 8 | 75.00 | 75.00 |
> | | 3 | 32 | 68.75 | 62.50 |
> || 4 | 96 | 60.42 | **68.75** |
> || 5 | 256 | 50.78 | **64.06** |
> || 6 | 640 | 52.34 | **55.16** |
> || 7 | 1488 | 45.30 | **52.69** |
> | **LLama3-70B** | 2 | 8 | 62.50 | 62.50 |
> || 3 | 32 | 68.75 | 65.63 |
> || 4 | 96 | 60.42 | **62.50** |
> || 5 | 256 | 56.25 | **57.42** |
> || 6 | 640 | 48.59 | **54.84** |
> || 7 | 1488 | 46.71 | **52.35** |
>
> Bold numbers in this table demonstrate the significant advantage of VoT particularly at higher difficulty levels. It indicates VoT's superior robustness in handling more complex spatial reasoning tasks.
> > **Weakness 3: Anthropomorphism**
>
> We apologize for some unnecessary anthropomorphism. We used *mental image* and *mind's eye* as metaphors to help readers intuitively understand the described mechanisms, without introducing unfamiliar terminology. We'll revise unnecessary usage which hinders scientific rigors.
> > **Question**
>
> Both the moving house and person icon are among the 30 symbols LLMs automatically generate to track the navigation progress, as described in the appendix E. In figure 8b, since the direction of each instruction is correct, the left example is a correct solution with imperfect intermediate visualization. Due to the limitation of this inherent visualization ability, even for the most advanced models, it's still challenging to generate a perfect visualization. We provide analysis about the accuracy of the final visualization in Table 2, which are lower than 30% in both visual tasks.
>
> We hope these clarifications will address the concerns raised and provide a clearer understanding of our work.
> ## References ##
> [1] Alan Baddeley. Working memory. Science, 255(5044):556–559, 1992

---

> > ### Comment · Reviewer_yVwi · 2024-08-09
> > **Thank you for the rebuttal**
> >
> > Thank you for the thoughtful rebuttal.  This cleared up some aspects of the method and results for me -- I hope you will include all of these clarifications in a revised version of the paper.

---

> > > ### Author Response · Authors · 2024-08-11
> > >
> > > We sincerely appreciate your feedback on our rebuttal. We're glad to hear that our responses helped clarify aspects of our methodology and findings. We will certainly incorporate all the clarifications provided in the rebuttal into a revised version of the paper to ensure it is as clear and comprehensive as possible for all readers.
> > >
> > > Your comments have been instrumental in improving the quality and clarity of our manuscript. If you have any remaining concerns, please let us know, and we will be more than willing to address them.

---

### Official Review · Reviewer_us5D · 2024-07-12

**Soundness:** 3
**Presentation:** 3
**Contribution:** 2
**Rating:** 5
**Confidence:** 4

**Summary:**

This paper focuses on enhancing the spatial reasoning capabilities of Large Language Models (LLMs) by introducing Visualization-of-Thought (VoT) prompting. Inspired by the human cognitive ability to visualize unseen objects - a process known as the Mind’s Eye - VoT visualizes reasoning processes to guide LLMs in tasks like natural language navigation and visual navigation in 2D grid worlds.

**Strengths:**

1. The introduction of Visualization-of-Thought (VoT) prompting represents a new method that mimics human cognitive processes, specifically the ability to visualize mental images for better spatial reasoning.
2. To investigate the effectiveness of the proposed VoT, this paper considers three spatial-related tasks, including natural language navigation, visual navigation, and visual tiling.

**Weaknesses:**

**Method

Some descriptions are unclear.

1. The method for obtaining the visuospatial sketchpad $v$ remains unclear. This paper describes $v$ as representing a visualization sequence in text form, but it does not specify how visual information (maybe images) is converted into text.

2. If visual information is transformed into text form, how does GPT-4V use it? Is it used as a textual representation of the visual image as well?

**Experiment

1. This paper also introduces the concept of state $s$. What is the performance comparison when this state is included versus when it is excluded?

2. In Table 1, for the visual navigation and visual tiling tasks, why does GPT-4V CoT perform the poorest, even yielding results lower than GPT-4 without visual information?

3. It seems that Section 5.2 aims to discuss “how visualization enhance final answers”, yet the connection between the results in Table 2 and this discussion remains unclear. It would be beneficial to include more detailed descriptions or results related to this.

4. In Section 5.3, Table 3 shows that Llama3-8B with VoT underperforms in most results compared to its counterpart with CoT. Does this suggest that the proposed VoT does not benefit less powerful language models?

**Questions:**

Please refer to weaknesses.

**Limitations:**

This paper provides the limitations.

---

> ### Author Rebuttal · Authors · 2024-08-06
>
> Thank you for your insightful comments, which are invaluable in helping us refine and clarify our work. We are grateful for the opportunity to address the points raised and provide further clarification on the aspects that may have been misunderstood.
> > **Weakness 1: Unclear descriptions**
>
> There might be some misunderstanding about the modality of *mental images* in LLMs during spatial reasoning. As explained in **line 42-44**, VoT leverages LLMs' potential ability of ascii-art for visualization, rather than depending on external visualization tools. We also provide explanations about ascii-art contained in code comments and some examples in appendix C. Regarding the input format, we specify the use of a 2D grid with special characters in **line 48**. Due to space limitations, detailed implementation information for the textual 2D grid dataset is presented in Appendix A, along with the rendering method used to generate the corresponding image input for GPT-4V.
>
> **In summary**, during spatial reasoning, LLMs generate ascii-art as text-form visualizations. The image input used by GPT-4V is produced by rendering the special characters from the textual 2D grid, rather than the reverse process.
>
> > **Weakness 2: Experiment**
>
> > **W2-1: Concept of state `s`**
>
> We introduce the concept of state `s` in line 139 to explain the **visual state tracking** nature of VoT. We discuss the visual state tracking behaviors among different baselines in **section 5.1**. As could be seen in Figure 5, there is a notable disparity in state tracking rate between **GPT-4 VoT** and **GPT-4 w/o Viz** across all tasks. We hope this comparison could address your concern.
>
> > **W2-2: Poor performance of GPT-4V**
>
> Your observation is indeed insightful. We were also suprised by the poor performance of GPT-4V, particularly in visual navigation task (being the poorest). After conducting a qualitative analysis of the responses, we hypothesize that visual hallucination may be hindering GPT-4V's performance.
>
> To further investigate, we conducted an additional experiment using 496 maps from the route planning task, focusing on a simple task: identifying the direction of the roads connected to the home icon. Despite the task's simplicity, where only one road begins at the home icon, both GPT-4 and GPT-4V occasionally provided multiple answers. Accuracy is measured by the ratio when the exact answer is outputed.
>
> | Setting | Accuracy | Recall | Avg answers output|
> | -------- | -------- | -------- | -------- |
> | GPT-4V | 2.02% | 66.13% | 2.32 |
> | GPT-4 | 34.07% | 76.81% | 1.50 |
>
> As could be observed in the table, GPT-4V with image input performs poorest in this easiest task. It indicates that GPT-4V suffers from hallucination in visual navigation task.
> > **W2-3: Unanswered question of section 5.2**
>
> We appreciate your observation regarding the clarity of Section 5.2. We did analyze the correlation between accurate visualization and final answers. We provided the metric explanation in **line 230-231**, which corresponds to the last column in table 2. And we concluded in **line 235** that accurate visualizations ensure a correct final answer with high probability (above 65%). Similarly, other metrics demonstrated in table 2 are also explained in the first paragraph of section 5.2. They measure the visualization capability of LLMs, e.g., to what degree LLMs can generate accurate visualizations.
>
> We recognize that the connection between the table and the narrative may not have been as explicit as intended. If these explanations address your questions, we'd like to adjust highlighting of numbers and observations to achieve a better understanding of table 2.
> > **W2-4 Less powerful LMs**
>
> We acknowledge that the overall performance of Llama3-8B is less impressive when using VoT, as noted in our discussion of limitations in **line 314-315**. We believe this can be attributed to the increased difficulty in handling complex tasks, leading to outcomes akin to random guessing in less powerful models. To further explore this issue, we conducted an analysis of performance trajectory in next-step-prediction task across various difficulty levels, as presented in **line 640-643 in appendix D**. However, as highlighted in **figure 9**, there's a modest improvement when the task complexity is lower (K ≤ 3). This suggests that VoT can potentially **benefit less powerful models in less complex scenarios**.
>
> We look forward to your further feedback and thank you for your valuable contributions to improving our manuscript.

---

> > ### Comment · Reviewer_us5D · 2024-08-13
> >
> > Thanks for your detailed response. It has addressed my concerns, I will raise the score to reflect this.

---

### Official Review · Reviewer_MbkM · 2024-07-13

**Soundness:** 4
**Presentation:** 4
**Contribution:** 3
**Rating:** 6
**Confidence:** 5

**Summary:**

The paper presents a novel approach called Visualization-of-Thought (VoT) prompting, aimed at enhancing the spatial reasoning abilities of Large Language Models (LLMs). Inspired by the human cognitive process known as the “Mind’s Eye,” the authors propose a method where LLMs visualize their reasoning steps to guide subsequent reasoning. The paper evaluates VoT across three spatial reasoning tasks: natural language navigation, visual navigation, and visual tiling. Experimental results demonstrate significant performance improvements with VoT over conventional prompting methods and existing multimodal large language models (MLLMs).

**Strengths:**

Strengths

1. **Innovative Direction**: The concept of eliciting spatial reasoning through visualization is novel and well-motivated. Drawing parallels with human cognition adds a strong cognitive science foundation to the work.

2. **Comprehensive Evaluation and Improved Performance**: The paper evaluates VoT across multiple tasks and benchmarks, providing a thorough analysis of its effectiveness. The results show that VoT significantly enhances spatial reasoning abilities in LLMs, outperforming existing methods. Meawhile, this paper includes detailed qualitative and quantitative analyses, shedding light on the mechanisms behind VoT’s success and its limitations.

**Weaknesses:**

Weaknesses

1. **Limited Task Diversity**: While the selected tasks cover various aspects of spatial reasoning, the evaluation could be broadened to include more complex and varied real-world scenarios. In the main paper, we even didnt see more commonly adopted benchmark performance comparison. Include more varied and complex spatial reasoning tasks to better evaluate the generalizability and robustness of VoT.

2. **Generalizability**: The paper primarily focuses on GPT-4 and GPT-4V. It would be beneficial to see how VoT performs across a wider range of models, particularly less advanced ones. It would be great to see more evaluations of VoT on a wider range of models, including less advanced ones, to assess its applicability across different LLMs.

3. **Prompt Sensitivity**: The effectiveness of VoT appears to be sensitive to the specific wording of prompts, which may limit its robustness in practical applications. Investigate ways to reduce the sensitivity of VoT to specific prompt wording, making it more robust for practical use.

**Questions:**

See above Weakness part.

**Limitations:**

See above Weakness part.

---

> ### Author Rebuttal · Authors · 2024-08-06
>
> Thank you for your thorough evaluation and constructive comments of our work. We'd like to clarify:
> > **Weakness 1: Limited Task Diversity**
>
> We fully appreciate the commonly adopted benchmark such as bAbI[1], StepGame[2], SpartQA[3], SPARTUN[4] etc., which lay solid foundations for spatial reasoning based on text understanding. As explained in section 2 (line 73-77), our main concern is that as the development of LLMs and their exceling in linguistic and logic reasoning, it's challenging to provide an accurate measure of their spatial awareness with these tasks. So we carefully select tasks to cover various aspects of spatial reasoning and offer flexible difficulty controls. These visual tasks with grid input avoid LLM's linguistic and logic "short-cut", and the dynamic nature of navigation tasks requires LLMs to track the changing state in the environment. Despite these tasks are relatively simple to humans, they are challenging to current stage LLMs.
> > **Weakness 2: Generalizability**
>
> We evaluated VoT and other baselines in GPT-family models (GPT-4/4V/3.5 turbo) and LLama3 family (LLama3-8B-instruct/70B-instruct). The resuluts and corresponding analysis are provided in **section 5.3**. Although the overall performance improvement of VoT in less advanced models is not as significant as in advanced models (least significant for LLama3-8B), the improvement in easier levels are distinctive (**discussed in appendix D**). As difficulty level increases, less advanced models exhibit irregular performance trajectory for both CoT and VoT. This observation indicates their inherent weakness, i.e., reliance on random guessing for increased task difficulty.
> > **Weakness 3: Prompt Sensitivity**
>
> We adopted task-agnostic 0-shot prompting for VoT as it's the simplest approach to mimic corresponding human cognition in LLMs, and assure a fair comparison with other baselines. We only scratched the surface of this ability in LLMs and hope this initiative could possibly inspire others. For practical usage, users can add task-aware specifications about what to visualize and how to visualize, to force LLMs appropriately generate visualizations to guide subsequent reasoning. More fundamentally, this limitation could be addressed by instruction tuning with visual state tracking data. Due to the space limitation, we discuss the possibility that VoT might benefit from code pre-training in **appendix C (line 600-627)**.
>
>
> We hope that our clarifications address your concerns and provide a better understanding of our work.
>
> ## References ##
> [1] Weston, Jason, Antoine Bordes, Sumit Chopra, Alexander M. Rush, Bart Van Merriënboer, Armand Joulin, and Tomas Mikolov. "Towards ai-complete question answering: A set of prerequisite toy tasks." arXiv preprint arXiv:1502.05698 (2015).
>
> [2] Shi, Zhengxiang, Qiang Zhang, and Aldo Lipani. "Stepgame: A new benchmark for robust multi-hop spatial reasoning in texts." In Proceedings of the AAAI conference on artificial intelligence, vol. 36, no. 10, pp. 11321-11329. 2022.
>
> [3] Mirzaee, Roshanak, Hossein Rajaby Faghihi, Qiang Ning, and Parisa Kordjmashidi. "Spartqa:: A textual question answering benchmark for spatial reasoning." arXiv preprint arXiv:2104.05832 (2021).
>
> [4] Mirzaee, Roshanak, and Parisa Kordjamshidi. "Transfer learning with synthetic corpora for spatial role labeling and reasoning." arXiv preprint arXiv:2210.16952 (2022).

---

### Decision · Program_Chairs · 2024-09-25

**Decision:**

Accept (poster)

**Comment:**

After the author rebuttal and the reviewer discussions, the paper received 2 borderline accepts, 1 weak accept, and 1 strong accept recommendations. The borderline reviewers mention that the author rebuttal clarified their concerns. The concerns included lacking technical details and additional experiments to confirm that the model scales.

The AC does not see a particular reason to turn down the opinions of the reviewers.